# Odor identity coding by distributed ensembles of neurons in the mouse olfactory cortex

Benjamin Roland[1], Thomas Deneux[2], Kevin M Franks[3], Brice Bathellier[2]*, Alexander Fleischmann[1]*

[1]Center for Interdisciplinary Research in Biology (CIRB), Collège de France, CNRS UMR 7241, INSERM U1050, Paris, France; [2]Unité de Neuroscience, Information et Complexité, Centre National de la Recherche Scientifique, UPR 3293, Gif-sur-Yvette, France; [3]Department of Neurobiology, Duke University, Durham, United States

**Abstract** Olfactory perception and behaviors critically depend on the ability to identify an odor across a wide range of concentrations. Here, we use calcium imaging to determine how odor identity is encoded in olfactory cortex. We find that, despite considerable trial-to-trial variability, odor identity can accurately be decoded from ensembles of co-active neurons that are distributed across piriform cortex without any apparent spatial organization. However, piriform response patterns change substantially over a 100-fold change in odor concentration, apparently degrading the population representation of odor identity. We show that this problem can be resolved by decoding odor identity from a subpopulation of concentration-invariant piriform neurons. These concentration-invariant neurons are overrepresented in piriform cortex but not in olfactory bulb mitral and tufted cells. We therefore propose that distinct perceptual features of odors are encoded in independent subnetworks of neurons in the olfactory cortex.

*For correspondence: bathellier@ unic.cnrs-gif.fr (BB); alexander. fleischmann@college-de-france.fr (AF)

**Competing interests:** The authors declare that no competing interests exist.

## Introduction

All sensory systems must be able to unambiguously determine stimulus identity in the face of variable stimulus intensity. In vision, for example, the perception of colors is stable across a wide range of luminance, despite the strong dependency of photoreceptor activation on wavelength and light intensity (*Nunn et al., 1984*). This problem is of particular importance in olfaction, given the massive and rapid fluctuations in odorant concentration, for example encountered in odor plumes in the environment (*Murlis and Jones, 1981*).

Molecular features of odorants are detected by odorant receptors. Odorant receptors are expressed on the dendrites of sensory neurons in the olfactory epithelium, and odorant receptors are broadly tuned such that each odorant typically activates multiple receptors (*Jiang et al., 2015*; *Malnic et al., 1999*). A given olfactory sensory neuron expresses one of a large repertoire of odorant receptors (*Buck and Axel, 1991*; *Zhang and Firestein, 2002*), and neurons expressing a given receptor project to two spatially invariant glomeruli in the olfactory bulb (*Mombaerts, 2001*). Thus, the molecular features of an odorant are represented as a discrete and stereotyped map of glomerular activity (*Bozza et al., 2004*; *Ma et al., 2012*; *Soucy et al., 2009*; *Uchida et al., 2000*). Odor information encoded in patterns of glomerular activity must then be integrated at higher olfactory centers in the brain to generate unified odor objects, defined by perceptual features such as odor identity and intensity (*Gottfried, 2010*; *Wilson and Sullivan, 2011*; *Wojcik and Sirotin, 2014*). The piriform cortex has been suggested to serve as such a site of integration.

The piriform cortex is a simple, three-layered paleocortical structure, which receives dense projections from mitral and tufted cells, the main output neurons of the olfactory bulb. Mitral and tufted cells extend an apical dendrite into a single glomerulus, and thus only receive direct excitatory input from sensory neurons expressing a single odorant receptor. Odor information encoded in the spatio-temporal patterns of mitral and tufted cell activity is then transmitted to higher olfactory centers, including the piriform cortex. Mitral and tufted cells project axons to large areas of the piriform cortex, without identifiable topographic organization (*Ghosh et al., 2011*; *Igarashi et al., 2012*; *Nagayama et al., 2010*; *Sosulski et al., 2011*). Individual piriform neurons receive inputs from multiple and broadly distributed glomeruli, thus providing an opportunity for molecular feature integration (*Apicella et al., 2010*; *Miyamichi et al., 2011*). Consistent with this model, optical stimulation of the olfactory bulb suggests that piriform neurons respond to combinations of co-active glomeruli (*Davison and Ehlers, 2011*; *Haddad et al., 2013*). Calcium imaging and electrophysiological recordings show that odors activate sparse ensembles of piriform neurons, which are distributed across the piriform cortex without apparent spatial organization (*Poo and Isaacson, 2009*; *Rennaker et al., 2007*; *Stettler and Axel, 2009*; *Tantirigama et al., 2017*). However, how information about the identity and intensity of an odor is encoded in the response patterns of piriform ensemble activity remains poorly understood.

The ability to unambiguously identify odors across a wide range of concentrations is essential for olfactory perception and behavior (*Cleland et al., 2011*; *Stopfer et al., 2003*; *Wojcik and Sirotin, 2014*). Indeed, rats can identify odorants with consistently high accuracy over a greater than 50,000-fold range in concentration (*Homma et al., 2009*). However, the specificity of odorant - receptor binding steeply depends on odorant concentration (*Jiang et al., 2015*; *Malnic et al., 1999*), and consequently, patterns of odor-evoked glomerular activity change with changing odorant concentration (*Bozza et al., 2004*; *Meister and Bonhoeffer, 2001*; *Rubin and Katz, 1999*; *Spors and Grinvald, 2002*). Similarly, the spatio-temporal patterns of mitral and tufted cell strongly depend on odorant concentration (*Banerjee et al., 2015*; *Bathellier et al., 2008*; *Fukunaga et al., 2012*; *Kato et al., 2013*; *Miyamichi et al., 2013*; *Sirotin et al., 2015*). Therefore, we sought to determine if the piriform cortex is capable of forming concentration-invariant representations of odor identity from sensory input that is concentration-dependent.

Here, we have used in vivo two-photon calcium imaging in anesthetized mice to record odor responses from large, unbiased ensembles of piriform neurons. We find that odor identity can accurately be decoded from the spatial patterns of local piriform odor responses. However, we also observe that piriform odor representations change substantially across a 100-fold range in odor concentration, degrading information about the identity of the odor. We propose a solution for this potential confound by identifying a subpopulation of concentration-invariant piriform neurons, which accurately encodes odor identity across a broad range of odor concentrations. These concentration-invariant neurons are present at numbers significantly above chance in piriform ensembles but not in olfactory bulb mitral and tufted cells, indicating that the ability to form concentration-independent representations of odor identity in functionally distinct neural subnetworks is an emergent property of piriform cortex.

## Results

### Representations of odor identity in the piriform cortex

We stereotaxically injected adeno-associated virus (AAV) expressing the calcium indicator GCaMP6s (*Chen et al., 2013*) into the piriform cortex of adult (6- to 10-week-old) mice. Ten days after infection, we surgically exposed the piriform cortex for two-photon imaging under ketamine/xylazine anesthesia. Viral expression of GCaMP6s resulted in dense labeling of layer II piriform neurons (*Figure 1a and b*, and Materials and methods). To monitor calcium signals in such large and densely packed neural ensembles, we developed an automated cell segmentation algorithm based on calcium signal similarities (*Figure 1—figure supplement 1* and Materials and methods). This algorithm operates on the entire data set obtained from individual imaging sites and efficiently identifies individual neurons by iteratively clustering neighboring pixels with high-signal covariance. Under these conditions, we could simultaneously monitor the activity of 100–400 (mean ± SD: 242 ± 105) neurons per imaging site.

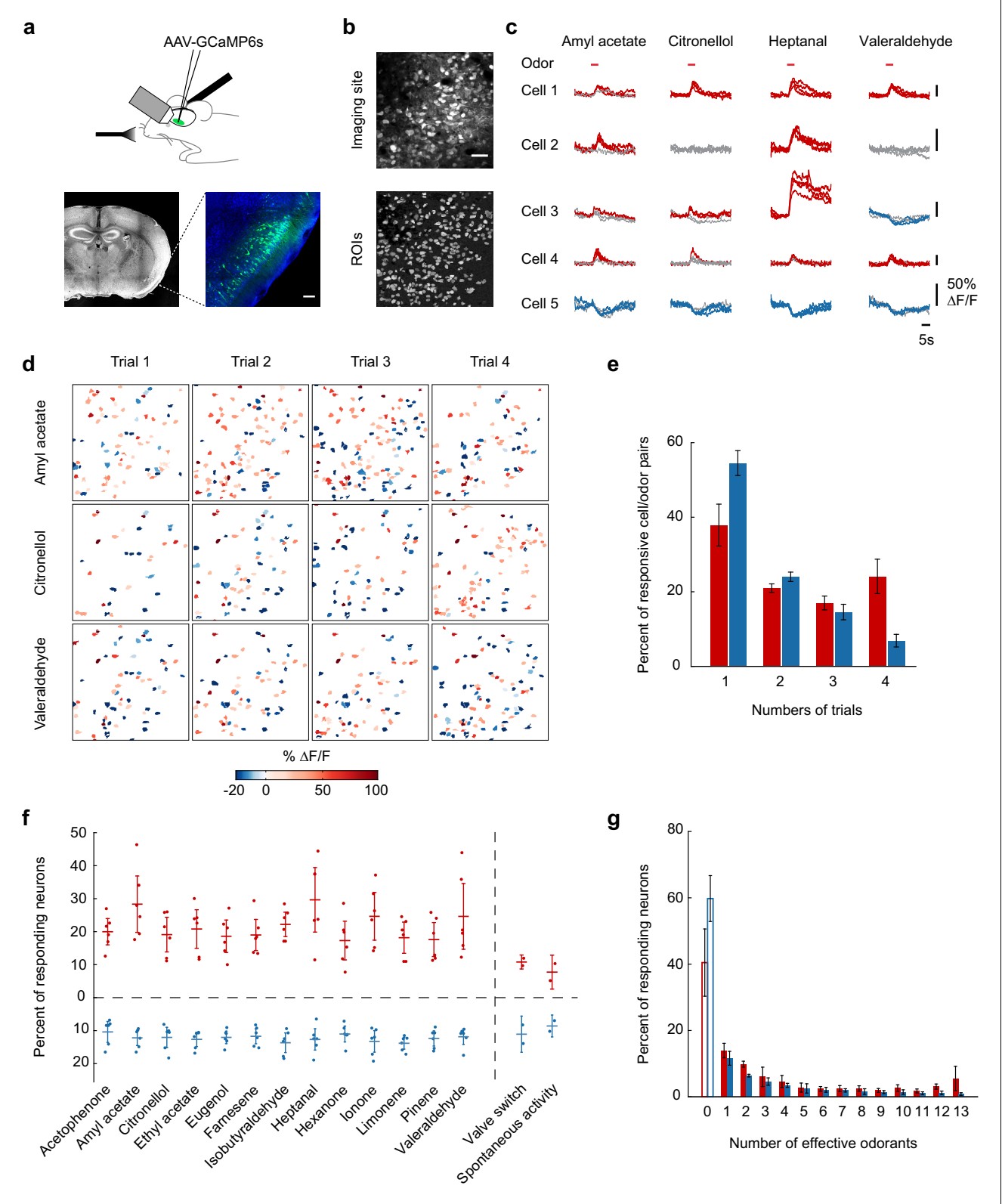

**Figure 1.** Calcium imaging of odor-evoked activity in piriform cortex. (a) (Top) Schematic of the experimental protocol. AAV-GCaMP6s was stereotaxically injected into piriform cortex. After 10 days, piriform cortex was surgically exposed and neural activity in response to odors was recorded with two-photon imaging. (Bottom) Coronal section of a brain used for piriform imaging. Scale bar: 100 µm. (b) (Top) Stack average of an imaging site, and (bottom) masks of the regions of interest (ROIs) identified by our clustering algorithm (see also *Figure 1—figure supplement 1*). Scale bar: 50 µm.
*Figure 1 continued on next page*

*Figure 1 continued*

(c) Single-trial example traces of four different cells (rows) in response to four different monomolecular odorants (columns). Cells respond with an increase (red traces) or a decrease (blue traces) in fluorescence (gray traces: non responsive trials). Note the different scale of ΔF/F values (y axis) for each neuron. Red bar: odor presentation. (d) Spatial patterns of piriform activity in response to four trials (columns) of three different monomolecular odorants (rows). ΔF/F values are clipped at 100% here and henceforth in all figures. Odorants activate sparse, distributed, and partially overlapping ensembles of piriform neurons. (e) Reliability of activation (red) and suppression (blue), measured as the number of trials each cell-odor pair responded to a given odorant, averaged across imaging sites (n = 6 sites in three mice). Error bars: 95% CI of the mean. (f) Percent of neurons activated (red) or suppressed (blue), averaged across four trials. Dots: single data points from individual imaging sites. Horizontal bars: mean across imaging sites (n = 6). Error bars: 95% CI of the mean. (g) Tuning curve of activation (red) and suppression (blue), averaged across imaging sites (n = 6). Error bars: 95% CI of the mean.

The following figure supplement is available for figure 1:

**Figure supplement 1.** Automated cell segmentation.

In initial experiments, to establish population coding properties of large ensembles of piriform neurons, we measured responses to a test panel of 13 different monomolecular odorants. Two-second odor pulses of these stimuli (1:10,000 dilution in mineral oil, 0.01% vol./vol.) elicited relatively sparse but partially overlapping activity of piriform neurons, consistent with one previously published report (*Stettler and Axel, 2009*) (*Figure 1c and d*). We observed that individual neurons responded selectively with an increase or decrease in fluorescence. Across six imaging sites in three mice (total number of neurons = 1706), 20% (±8.4%) of the neurons responded with an increase in fluorescence, and 11% (±5.7%) of the neurons responded with a decrease in fluorescence (*Figure 1f*). Most neurons responding with an increase in fluorescence exhibited narrow stimulus tuning, with the exception of a small subpopulation of neurons (8.5 ± 5.4%) that responded to 12 or all 13 odorants of the test panel. Neurons responding with a decrease in fluorescence exhibited similarly selective odor tuning, but only a minimal number of broadly tuned neurons could be observed (*Figure 1g*). Strikingly, many neurons displayed high trial-to-trial variability in response to the repeated delivery of the same odorant (*Figure 1e*).

We next sought to quantify the similarity of population responses evoked by different odorants. For each trial, we constructed population activity vectors, defined as the mean temporally deconvolved change in fluorescence of all simultaneously recorded cells of an imaging site over a 4 s time window after stimulus onset (*Figure 2a*). We then computed pairwise cross-correlations between all single-trial population activity vectors. The results of this analysis for the imaging site shown in *Figure 1b* are displayed in *Figure 2b* as a cross-correlation matrix. As indicated by the square boxes along the diagonal (4 × 4 trials), repeated exposure to the same odorant triggered highly correlated population response patterns (intra-odorant cross-trial correlation coefficients, neurons pooled across six imaging sites: 0.67 ± 0.07 (across odorants)). However, this cross-correlation analysis also revealed that patterns elicited by different odorants were fairly similar (mean inter-odorant correlation coefficient: 0.44 ± 0.08). Such overlap may, at least in part, be a consequence of correlated noise at a given imaging site. We therefore pooled neurons across imaging sites and projected this pseudo-population onto the first three principle components in principal component space (*Figure 2c*). While response patterns for a few odorants appeared to segregate from each other, substantial overlap remained between neural ensembles encoding different odorants.

This raises the question as to whether the spatial patterns of odor-evoked piriform activity we can observe with calcium imaging, which lacks precise temporal information at small time scales, contain sufficient information to discriminate between different odorants. To address this question, we tested the accuracy with which a linear classifier correctly identify odorants based on single-trial response patterns (see Materials and methods). Despite both trial-to-trial variability and considerable overlap in response patterns to different odorants, the classifier could correctly identify all 13 stimuli in our odorant test panel (*Figure 2d*). Classification accuracy reached 94% when pooling all six imaging sites (*Figure 2e*). For individual imaging sites, the average classification accuracy was 71% (±5.8%) (*Figure 2—figure supplement 1*). Correct classification slowly rises after odor onset to reach high accuracy within 0.5 s. This relatively slow rise can largely be explained by the slow rise time of the calcium indicator GCAMP6s (*Chen et al., 2013*). Interestingly, classification accuracy

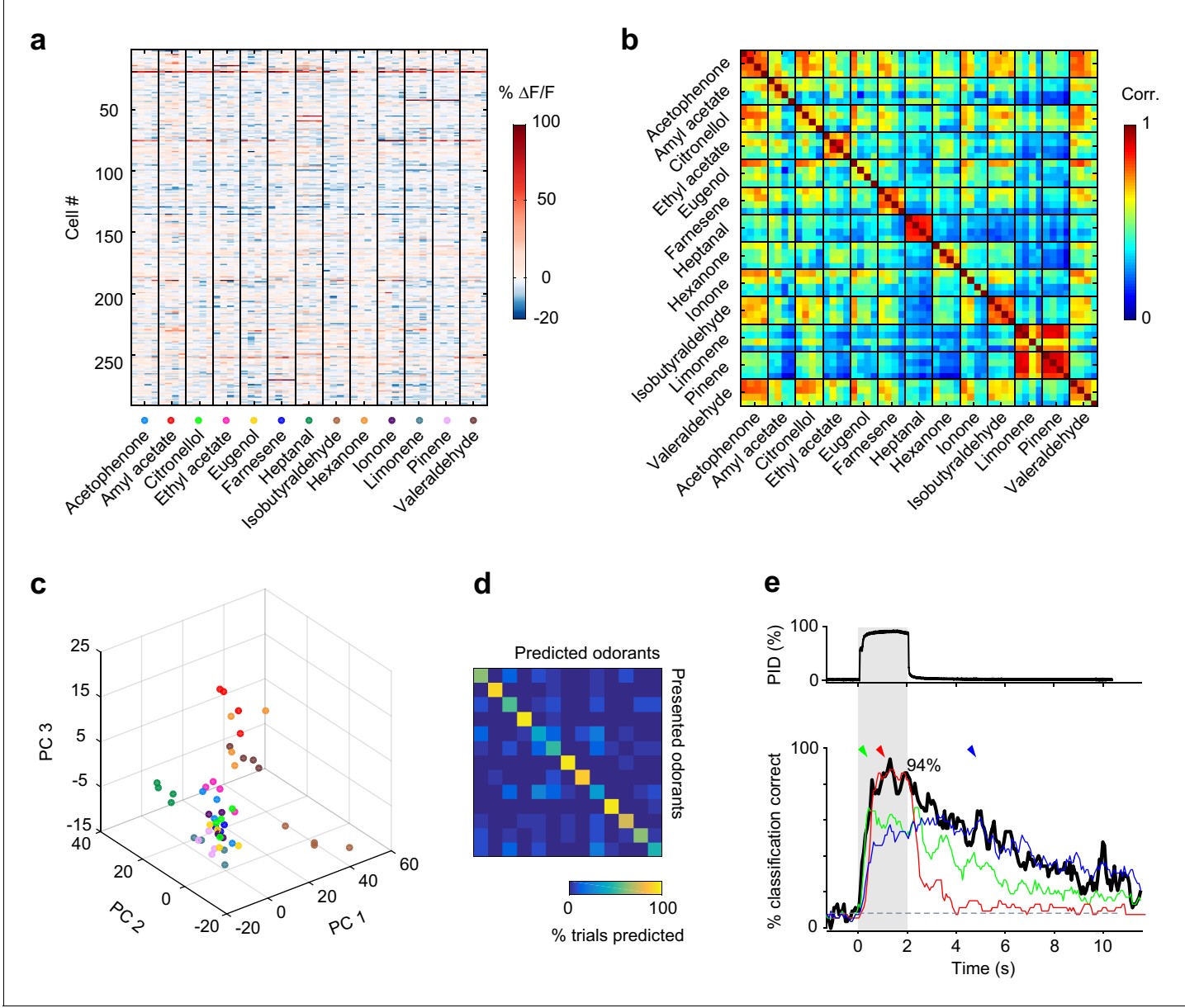

**Figure 2.** Odor identity is encoded in the spatial patterns of piriform activity. (a) Population response of the imaging site shown in *Figure 1a*. The mean ΔF/F value after stimulus onset for each trial (columns) in each cell (rows) is color-coded. (b) Similarity matrix obtained by computing the pairwise correlation coefficients between all population response vectors in (a). Squares along the diagonal (4 × 4 trials) represent the similarity of responses to four exposures of the same odorant (intra-stimulus cross-trial correlations). (c) Patterns of piriform activity in response to single odor presentations (dots represent different odorants, color-coded as in (a)) projected onto space of the first three principal components (accounting for ~20% of the total variance). Neurons were pooled across imaging sites. (d) Confusion matrix summarizing the performance of a linear classifier trained to discriminate the odorants in A, summed over imaging sites (n = 6 sites in three mice). (e) Percent of trials correctly identified by a linear classifier, for neurons pooled across all six imaging sites (blackline). Green, red and blue lines represent classification accuracy obtained with a classifier trained on response patterns at 0.27 (green arrowhead), 1 (red arrowhead) and 3 (blue arrowhead) s after odor onset. Gray dashed line: theoretical chance level. Gray square: odor exposure.

The following figure supplement is available for figure 2:

**Figure supplement 1.** Classification accuracy is consistent across mice and imaging sites.

remained significantly above chance level for several seconds after odor delivery, despite a marked decline in odor-evoked fluorescence (*Figure 2e*). This finding is consistent with the observation that odor-specific network configurations persist for extended periods of time in the olfactory bulb (*Bathellier et al., 2008*; *Friedrich and Laurent, 2001*). To determine whether this sustained piriform activity is odor-specific, we trained the classifier at defined time points after odor onset (0.27, 1, and 3 s, *Figure 2e*, colored arrows) and measured classification accuracy over time. We observed that a classifier trained on response patterns 1 s after odor onset accurately predicted odor identity during odor delivery, but classification accuracy readily declined after odor offset. In contrast, a classifier based on response patterns at 0.27 and 3 s after odor onset yielded a lower classification accuracy; however, classification success was more stable over time. This suggests that early and late (post-odor offset) representations are based on weak activity configurations present during odor exposure but masked by another much stronger component, which quickly vanishes after the odor is withdrawn. Thus, odor representations are dynamically rearranged after odor offset to maintain odor information for several seconds.

Previous anatomical and functional experiments suggested that odor information was encoded in randomly distributed ensembles of piriform neurons (*Ghosh et al., 2011*; *Illig and Haberly, 2003*; *Sosulski et al., 2011*; *Stettler and Axel, 2009*). Therefore, to test whether odorant-selective piriform neurons were clustered or randomly distributed across individual imaging sites, we selected cells that were significantly modulated by odorants (1-way ANOVA, $p < 0.05$) and mapped their odorant preference (i.e. the odorant that triggered the strongest response, see example map in *Figure 3a*). As a sensitive measure of spatial clustering, we computed for each imaging site a nearest neighbor index (NNI *Theodoridis and Koutroumbas, 2009*), as the mean distance of a cell preferring a given odor to the nearest cell preferring the same odor. We then computed the NNI distribution for 1000 spatially shuffled maps to estimate the probability p that the observed value can be explained by random spatial organization (null hypothesis). Using this measure, we found no evidence for spatial clustering in any of the six imaging sites (p values of 0.49, 0.81, 0.09, 0.21, 0.91, 0.79). To validate the sensitivity of this test to detect spatial clusters, we again simulated populations but now with subtle inhomogeneities in the spatial distributions of neurons. We divided the simulated site into 16 equally sized sub-areas and allowed preference for a given odorant to occur in only three or four randomly chosen sub-areas (*Figure 3b*). In this case, comparing observed and simulated NNI resulted in highly significant values for p.

These observations suggest that information about odor identity is homogeneously distributed across individual imaging sites. To further illustrate this idea, we sequentially defined each cell within an imaging site as a 'starter cell', and iteratively built neural ensembles of increasing size by adding neighboring neurons (*Figure 3c*). We observed that different clusters of the same size encoded odorant identity with similar accuracy, and that no 'hotspots' of classification were observed. Finally, we found that classification accuracy was very similar for the six different imaging sites in three mice, which were up to 1 mm apart along the rostro-caudal axis of the piriform cortex (*Figure 2—figure supplement 1*, and data not shown).

Taken together, our data show that odorants activate spatially distributed ensembles of piriform neurons with significant overlap and variability, and no apparent spatial organization. Furthermore, the spatial patterns of odor-evoked piriform activity contain sufficient information to robustly decode stimulus identity.

## Patterns of piriform activity decorrelate with increasing concentrations

The ability to correctly classify individual trials as corresponding to a given odorant reveals that odor response patterns are different, but does not directly address how odor identity is encoded within piriform ensembles. Importantly, perceived odor identity of monomolecular odorants typically remains stable across a large range of odor intensities (*Homma et al., 2009*), providing an experimental opportunity to more directly address this question. Therefore, to test if odor identity coding is stable across changing odorant concentrations, that is concentration-invariant, we next analyzed piriform neural activity in response to three different odorants over a 100-fold range in concentrations (acetophenone, ethyl acetate, and hexanone; at 1:10,000, 1:1,000, 1:100 vol./vol. dilution in mineral oil). We confirmed, using photoionization detector (PID) measurements, that odorant presentations were reliable and scaled according to volumetric ratios (*Figure 4—figure supplement 1*). We analyzed 13 imaging sites in 11 mice (total number of neurons = 2935). We found that the

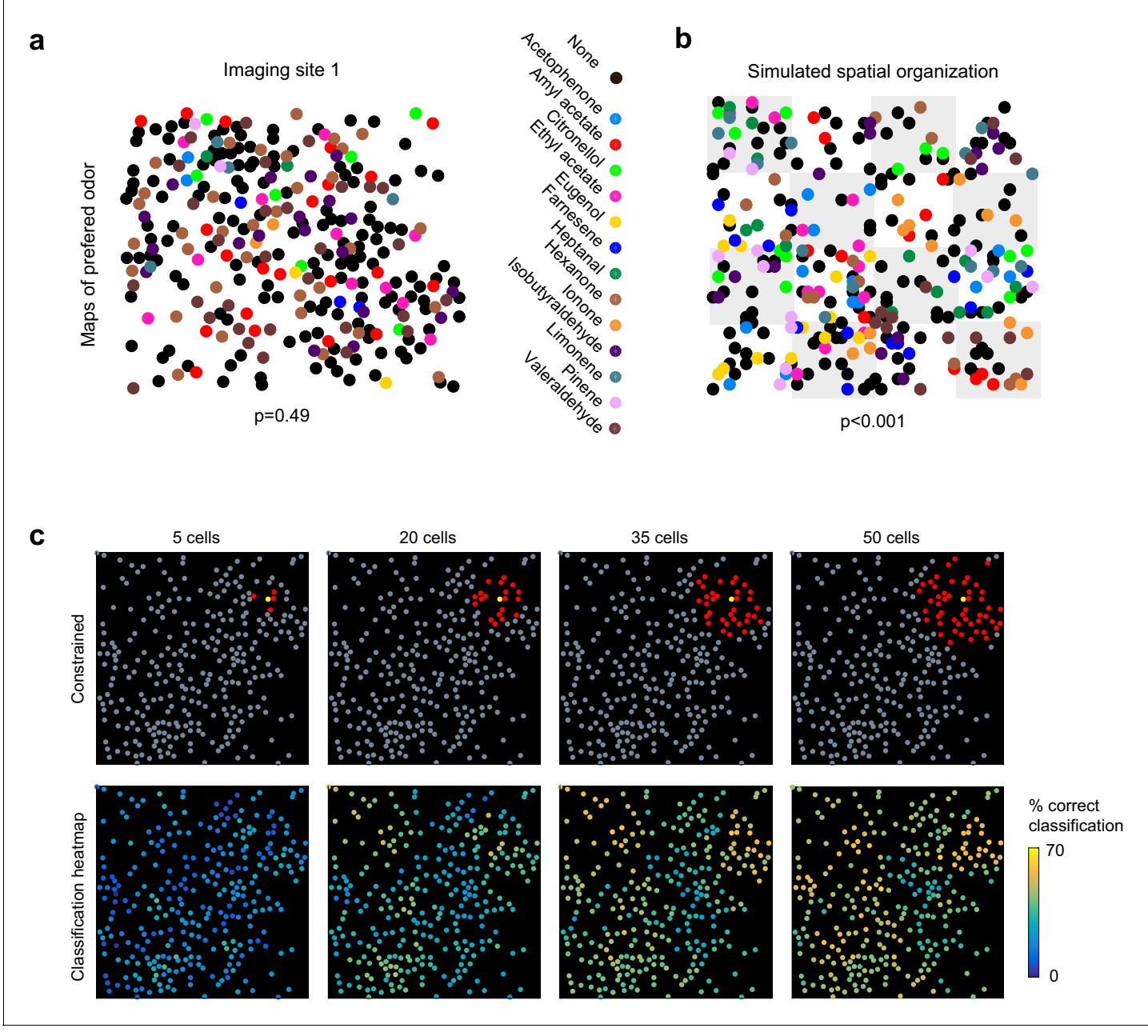

**Figure 3.** No evidence for topography in patterns of piriform activity. (a) Odorant preference map for an imaging site, with the odorant preference of individual piriform neurons color coded as in *Figure 2*. (b) Example of simulated clustering. The imaging site was divided into 16 equally sized sub-areas and preference for an odorant was allowed to occur in only three or four randomly chosen sub-areas. The p value indicates the probability that the computed nearest neighbor index is different from randomly distributed neurons. (c) (Top) Example of an ensemble of piriform neurons (in red) locally constrained around a 'starter cell' (in yellow), and used for classification of the stimulus set of *Figure 2*. (Bottom) Heatmaps of the classification accuracy of different starter cells for piriform ensembles of increasing size.

fraction of both activated and suppressed neurons increased moderately but statistically significantly with increasing concentrations (*Figure 4a and b*, mixed effect ANOVA, significant effect of concentration on the fraction of activated neurons: $F_{(2,96)} = 52.49$, $p<0.01$, and on the fraction of suppressed neurons: $F_{(2,96)} = 28.79$, $p<0.01$). Thus, population sparseness decreases with increasing concentrations. We then asked how changes in odorant concentration impacted the response properties of individual neurons. To follow the evolution of each cell's odorant selectivity, we calculated

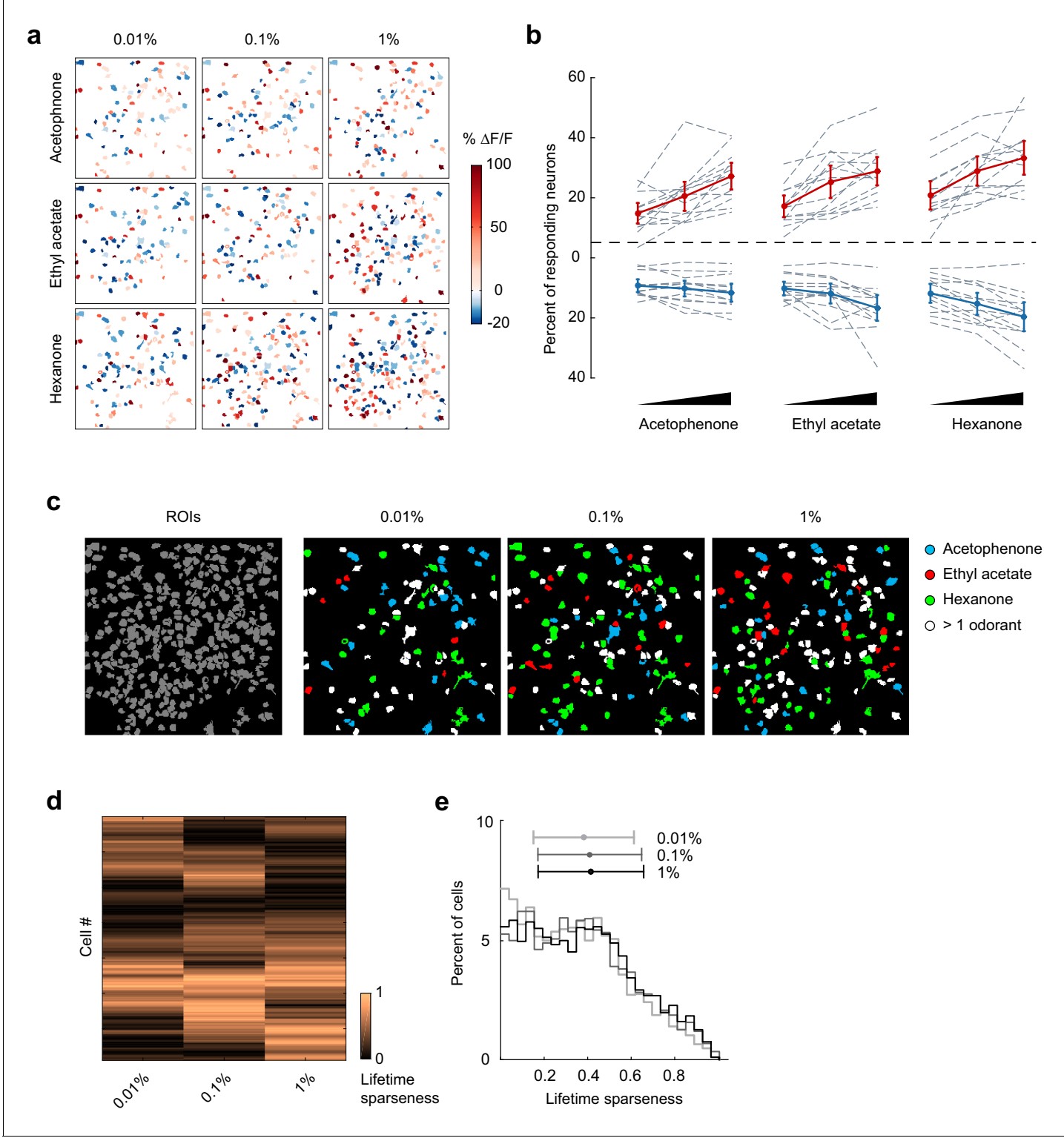

**Figure 4.** Odor-evoked activity and sparseness of individual piriform neurons depends on odorant concentration. (**a**) Spatial patterns of piriform activity in response to acetophenone, ethyl acetate and hexanone at three different concentrations (1:10,000, 1:1,000, 1:100 vol./vol. dilutions). (**b**) Percent of neurons activated or suppressed by acetophenone, ethyl acetate, and hexanone at three different concentrations (1:10,000, 1:1,000, 1:100 vol./vol. dilutions). Dashed gray lines represent individual imaging sites, thick red (activation) and blue (suppression) lines represent averages across sites (n = 13 sites in 11 mice). Error bars: 95% CI of the mean. (**c**) Spatial patterns of piriform activity in response to acetophenone, ethyl acetate and hexanone at three different concentrations (1:10,000, 1:1,000, 1:100 vol./vol. dilutions). Only cells responding at least two out of four trials are depicted. Cells

*Figure 4 continued on next page*

*Figure 4 continued*

responding to multiple odorants are color-coded in white. (**d**) Matrix of lifetime sparseness across concentrations for the cells in (**c**), sorted by hierarchical clustering. (**e**) Distribution of lifetime sparseness of all neurons pooled across imaging sites (n = 13, total number of neurons = 2935).

The following figure supplement is available for figure 4:

**Figure supplement 1.** Odorant concentration scales with nominal dilution and breath period is independent of odorant concentration.

lifetime sparseness across the 100-fold range in concentration (see Materials and methods). We observed diverse concentration-induced changes in cell selectivity across the population, indicating that an increase in odorant concentration did not systematically broaden odor tuning (*Figure 4c and d*). Indeed, when considered across the entire population, the distribution of lifetime sparseness of individual neurons was maintained across stimulus intensities (mean ± SD: 0.01%: 0.35 ± 0.23, 0.1%: 0.37 ± 0.24, 1%: 0.38 ± 0.24, *Figure 4e*).

To visualize changes in the population response patterns with increasing odorant concentration, we next rearranged the population response matrix of the imaging site shown in *Figure 3a*, using hierarchical clustering (*Figure 5a and b*). This analysis further supported the observation that neural responses varied with the odorant stimulus, and that response magnitudes could increase or decrease with increasing odorant concentrations. To quantify the similarities of response patterns elicited by odorants at different concentrations, and to evaluate the contributions of these different response profiles to the encoding of the identity of an odorant, we next calculated the cross-trial correlations between individual response vectors of neurons pooled across 13 imaging sites (*Figure 5c*). We first noted that pair-wise trial correlations increased with concentration, indicating that trial-to-trial variability across the population decreases at higher concentrations (mean ± SD across odorants: 0.01%, 0.67 ± 0.03; 0.1%, 0.72 ± 0.06; 1%, 0.74 ± 0.05). Note that this decrease in trial-to-trial variability partly accounts for the increased number of cells with responses that are statistically significantly different from mineral oil at higher concentrations (*Figure 4b*). Furthermore, we found that responses to a given odorant at a given concentration were significantly more correlated than responses to a given odorant at different concentrations (mixed-effect ANOVA, effect of concentration $F_{2,96} = 51.39$, p<0.01). This concentration-dependent decorrelation was gradual: the average correlation between response patterns elicited by the three odorants at low (0.01% vol./vol.) and intermediate (0.1% vol./vol.) concentrations was 0.53 (±0.12), while the average correlation between response patterns at low and high (1% vol./vol.) odorant concentrations dropped to 0.36 (±0.12) (*Figure 5c and d*). Thus, the spatial patterns of piriform odor responses were not concentration-invariant. Instead, response patterns to a given odorant changed substantially as odor concentration increased and became as different over a 100-fold change in concentration as responses to a different odorant.

## A concentration-invariant subpopulation of piriform neurons

Because overall piriform patterns of piriform activity change across odorant concentrations, we next asked if we could identify subnetworks of concentration-invariant piriform neurons that can represent the identity of an odor independent of its intensity. We used a linear regression approach (*Rigotti et al., 2013*) to determine if individual neurons were present within piriform ensembles whose response profiles could be accounted for solely by the identity of the odorants, irrespective of their concentrations. We performed an analysis of variance of each cell's response profile, with odorant concentration and odorant identity as the two explanatory variables (*Figure 6a*). Consistent with our qualitative inspection of response profiles (*Figure 5a and b*), we found that 37.2 ± 11.7% of cells were selectively responsive to odorant, while 27.8 ± 10.6% of cells showed responses that were modulated by an interaction between odorant identity and concentration (*Figure 6a*). Crucially, of the cells that were selectively modulated by odorant, 30.1% ± 11.6%, constituting 10.4 ± 4.5% of the total population of cells, exhibited concentration-invariant responses for all three odors, according to the analysis of variance (*Figure 6c*). Representative response traces of concentration-invariant neurons are shown in *Figure 6b*. Thus, although many neurons displayed mixed selectivity to the

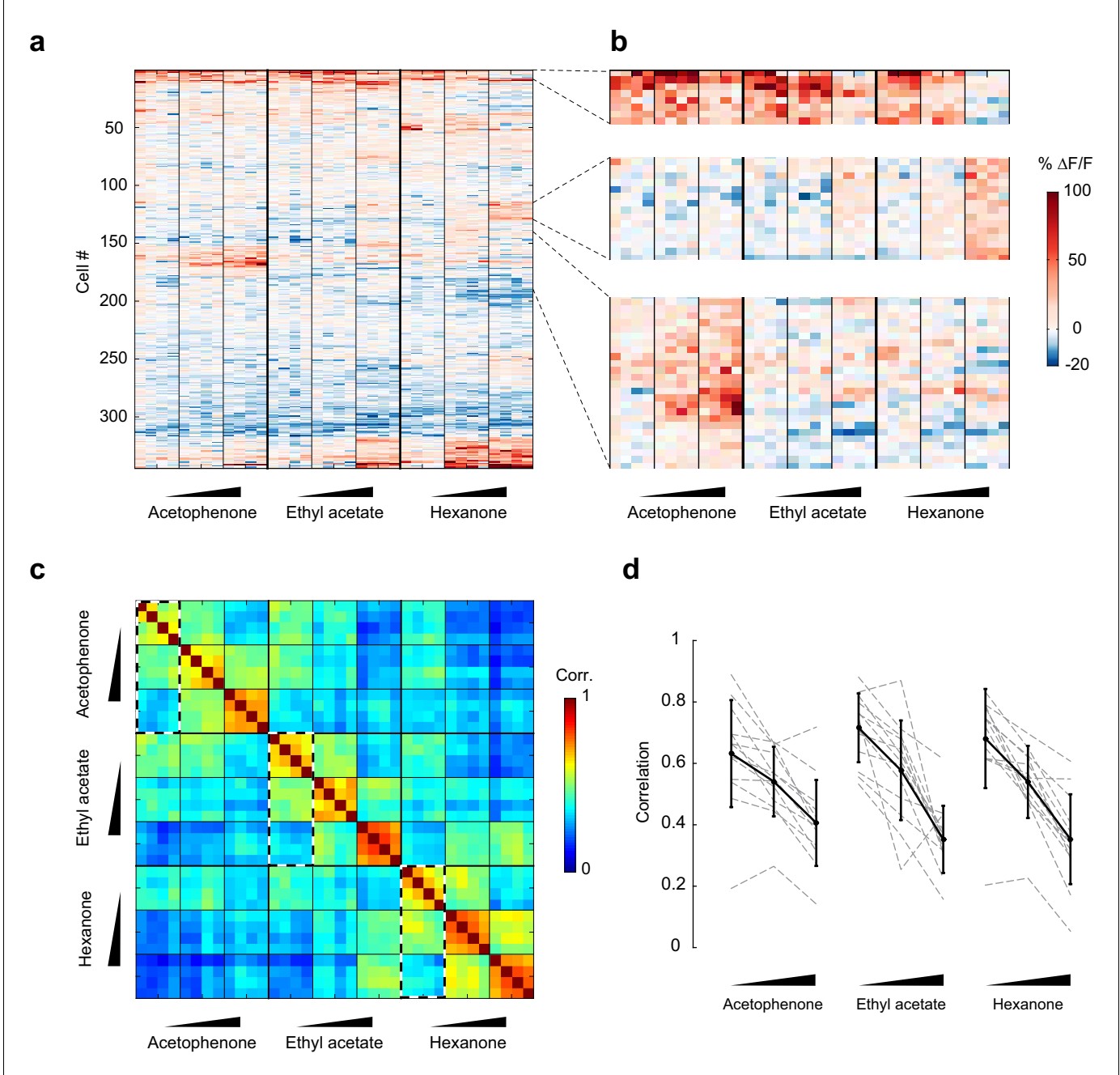

**Figure 5.** Patterns of piriform activity decorrelate with increasing odorant concentrations. (a) Population response of the imaging site in *Figure 3a* to acetophenone, ethyl acetate and hexanone at three different concentrations (1:10,000, 1:1,000, 1:100 vol./vol. dilutions). Cells are sorted by hierarchical clustering. (b) Example response profiles of cells suppressed at higher concentrations for all three odorants (top panel), cells moderately enhanced by increasing concentrations of ethyl acetate or hexanone (middle panel), and cells strongly enhanced by increasing concentrations of acetophenone (bottom). (c) Similarity matrix obtained by computing the pairwise correlation coefficients between all response vectors pooled across imaging sites. Squares along the diagonal (4 × 4 trials) represent the similarities of responses to a single odorant/concentration pair (intra-stimulus cross-trial correlations). Large squares (12 × 12 trials) represent the similarities of responses to an odorant at varying concentrations (intra-odorant cross-trial correlations). The similarities of responses to increasing concentrations of a given odorant (intra-odorant inter-concentrations similarity) are highlighted by dashed line rectangles. (d) Correlation coefficients of the patterns of piriform activity elicited at increasing concentrations with the patterns elicited at low concentrations. Dashed gray lines represent individual imaging sites, thick black lines the average across sites (n = 13 sites in 11 mice). Error bars: 95% CI of the mean. Patterns of piriform activity along increasing concentrations gradually decorrelate from the patterns elicited at low concentration.

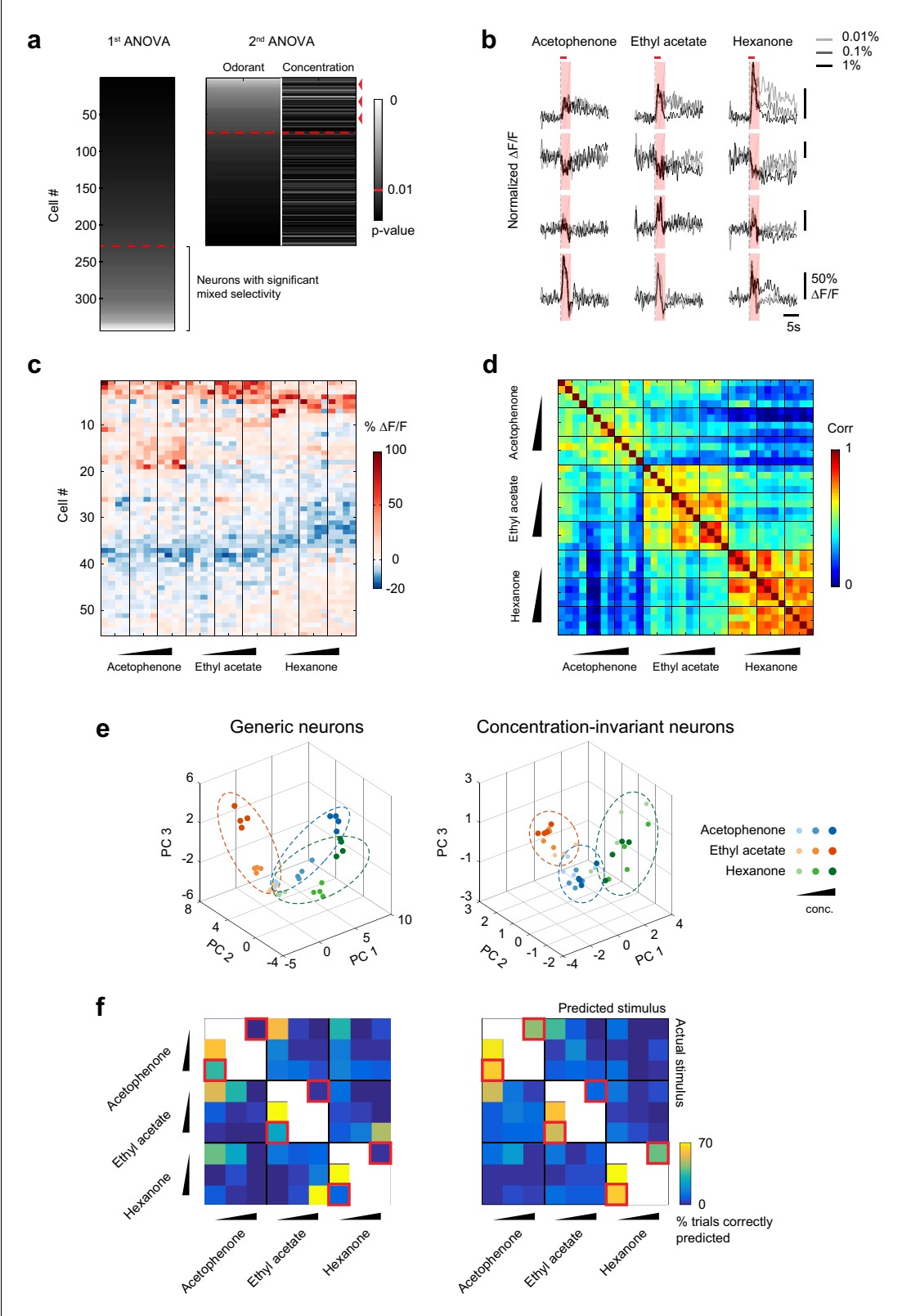

**Figure 6.** A concentration-invariant subnetwork of piriform neurons. (a) Summary of the analysis of variance of the response profiles of cells in *Figure 4a*. Two ANOVAs are performed successively. In the first step, neurons with mixed selectivity are excluded from the analysis. Concentration-invariant cells are then identified as cells significantly modulated by odorant identity but not concentration (red arrowheads: example cells). These cells are used for subsequent analyses against the population of all other 'generic neurons'. Red dashed line: significance threshold, set at p=0.01. (b)

*Figure 6 continued on next page*

*Figure 6 continued*

Deconvolved response traces of 4 concentration-invariant cells (rows) to acetophenone, ethyl acetate and hexanone at three different concentrations (light grey: 1:10,000, dark grey: 1:1,000, black 1:100 vol./vol. dilutions). ΔF/F values are normalized to each cell's maximum ΔF/F. Red bar: odor presentation. Shaded area: time interval used to integrate ΔF/F values for the analysis of variance in (a). (c) Response matrix of the population of cells in (a) exclusively modulated by odorant identity. ΔF/F values do not vary significantly with increasing odorant concentrations. (d) Similarity matrix obtained by computing the correlation coefficients between patterns of concentration-invariant neurons pooled across imaging sites (n = 13). Responses of concentration-invariant neurons to different concentrations of a given odorant (intra-odorant inter-concentrations similarity) are as correlated as responses to the same odorant/concentration pair. (e) Patterns of activity of generic neurons (left) or concentration-invariant neurons (right) in response to single-odor presentations (dots) projected onto space of the first three principal components. Neurons were pooled across imaging sites (n = 13). (f) Confusion matrix summarizing the accuracy of the classification of odorant identity by generic neurons (left) or concentration-invariant neurons (right) in a generalization task (see Materials and methods), summed over imaging sites (n = 13). The classifier assigns each odor trial to one of seven stimulus groups. Two concentrations of the tested odorant (including the tested concentration) are excluded from the training data (white boxes). Difficult generalization tasks across a 100-fold change in odorant concentration are highlighted by the red squares.

The following figure supplements are available for figure 6:

**Figure supplement 1.** Cross-correlation and principal component analysis of concentration-invariant piriform neurons.

**Figure supplement 2.** Response properties and spatial organization of concentration-invariant piriform neurons.

odorant identities and concentrations tested in our experiment, a substantial fraction of concentration-invariant neurons can be identified within piriform neural ensembles.

To verify our selection procedure for concentration-invariant piriform neurons, we quantified the similarities of their response patterns across the 100-fold odorant concentration range. We found that, for a given odorant, responses remained highly correlated across increasing concentrations (mean inter-concentration similarity ± SD across odorants, 0.01% to 0.1%: 0.60 ± 0.09; 0.01% to 1%: 0.57 ± 0.10, *Figure 6d*, and *Figure 6—figure supplement 1*). These across-concentration response correlations for concentration-invariant subpopulations approached those for within-concentration responses observed in the general population (0.65 ± 0.10), although a small but significant difference remained (Wilcoxon rank sum test, p<0.05).

To qualitatively display differences in the odor representation formed by the concentration-invariant subnetwork of neurons compared to the rest of the piriform ensemble, we next projected their population response patterns onto the first three principal components in principal component space. This analysis revealed that odorant representations of concentration-invariant neurons clustered irrespective of concentration, while odorant representations of generic neurons clustered by concentrations of the same odorants, but not systematically by odorant groups (*Figure 6e* and *Figure 6—figure supplement 2*). To more quantitatively evaluate the odor-coding properties of concentration-invariant piriform subnetworks and to test if concentration-invariant neurons could generalize odorant identity across changing odorant concentrations, we next trained a linear classifier to predict odorant identity based on a single concentration. We then tested the classifier on all other odorants and concentrations ('generalization learning', *Figure 6f*, see Materials and methods). We found that the classification accuracy of odorant identity across a 10-fold change in odorant concentration was similar between the concentration-invariant subpopulation of neurons and the entire population of 'generic' neurons. However, for the more difficult generalization tasks across a 100-fold change in odorant concentration (0.01% to 1%, and 1% to 0.01%, *Figure 5f*, red squared boxes), subnetworks of concentration-invariant neurons were much more accurate in predicting odorant identity than the entire population of 'generic' neurons (1% to 0.01% generalization: concentration-invariant cells mean = 63.5 ± 27.1%, Mann-Whitney test n = 13 U=20, Benjamini and Hochberg's FDR adjusted p<0.01, 'generic' neurons mean = 19.2%±17.8%, Mann-Whitney test n = 13 U=82, FDR adjusted p=0.39). Taken together, these analyses demonstrate that for a given set of odorants and concentrations it is possible to identify a subpopulation of piriform neurons that can encode the identity of an odor largely independent of odor intensity.

How stable are concentration-invariant piriform subnetworks with varying stimulus intensity range and complexity? To address this question, we first identified concentration-invariant neurons across a 10-fold instead of a 100-fold range in odorant concentration. Using the same selection criteria as

for the original data set, this analysis yielded 19.2% (±7.4% SD across experiments) concentration-invariant neurons. Second, we identified concentration-invariant neurons for pairs of two odorants instead of the three odorants in our test panel. We found that 22% (±8%) of the cells were concentration-invariant for at least one pair of odorants. Of those cells, 49% were identified as concentration-invariant for all three odorants while the response of the other 51% of cells was modulated by the concentration of one of the three odorants. Taken together, these data suggest that concentration-invariant subnetworks of neurons can be modulated by stimulus complexity and concentration, yet remain relatively stable within the range of our stimulus set.

We then tested if concentration-invariant neurons exhibited response profiles that differentiate them from other neurons. We compared response rise time (from 10% to 90% of peak $\Delta$F/F, concentration-invariant neurons: 0.55 ± 0.05 s; generic neurons: 0.52 ± 0.10 s) and duration (width at 50% peak $\Delta$F/F, concentration-invariant neurons: 2.02 ± 0.07 s; generic neurons: 2.01 ± 0.11 s), odor-evoked peak change in fluorescence (deconvolved $\Delta$F/F, concentration-invariant neurons: 10.0 ± 0.7%; generic neurons: 9.2 ± 1.3%), and trial-to-trial variability, but found no significance between the two populations (p>0.05, Wilcoxon ranked sum test, n = 13) (*Figure 6—figure supplement 1*). We next examined whether concentration-invariant cells were spatially clustered. Visual inspection of the localization of concentration-invariant cells at individual imaging sites did not reveal obvious clustering (*Figure 6—figure supplement 1*). Furthermore, performing the statistical analysis based on the nearest neighbor index (see *Figure 3*), we found that the organization of concentration-invariant neurons at 11 out of 13 imaging sites was undistinguishable from the random distribution obtained for shuffled data. In 2 of the 13 imaging sites, spatial distributions were moderately but significantly different from random.

Taken together, our analysis identifies a subpopulation of piriform neurons, with response profiles and spatial distributions that are similar to other odor-responsive neurons, but which encode odor identity independent of concentration.

## Concentration-invariant neurons are overrepresented in piriform cortex but not in the olfactory bulb

A representation of odor identity emerges in piriform cortex from the integration of odor-evoked mitral and tufted cell activity from the olfactory bulb. Considerable normalization of odor-evoked neural activity across a range of odorant concentrations has been observed in the olfactory bulb (*Banerjee et al., 2015*; *Kato et al., 2013*; *Miyamichi et al., 2013*; *Roland et al., 2016*; *Zhu et al., 2013*), suggesting that concentration-invariant piriform odor responses could be inherited from the olfactory bulb. Alternatively, the formation of segregated, concentration-invariant odor identity representations in subpopulations of piriform neurons may emerge within cortex itself. Therefore, to distinguish between these models, we next analyzed odor-evoked responses of olfactory bulb mitral and tufted cells. We used previously described mitral/tufted cell calcium imaging data (*Roland et al., 2016*) (*Figure 7a*), obtained under equivalent experimental conditions but using GCaMP3 instead of GCaMP6 as the calcium indicator. We then performed an analysis of variance of each cell's response profile, as described above for the piriform imaging data (19 imaging sites in eight mice, total number of cells = 523, *Figure 7b*).

Visual inspection of the response matrix suggested that the majority of mitral cell responses are concentration-dependent (*Figure 7c*). Consistent with this impression, our analysis of variance indicated that 25.9% ± 14.9% of neurons exhibited odorant-selective responses. Of these, only 20.5 ± 15.1%, constituting only 5.2 ± 5.9% of the total population were concentration-invariant. Thus, many fewer neurons were exclusively modulated by odorant identity in the olfactory bulb compared to the piriform cortex (*Figure 7d*) (olfactory bulb, 5.2 ± 5.9%, piriform cortex 10.4 ± 4.5%, Mann-Whitney test, $n_{ob}$ = 19, $n_{pir}$ = 13, U = 59, p<0.01). However, we imaged more piriform neurons (2935) than olfactory bulb mitral cells (523). Therefore, to ensure that differences in the fractions of concentration-invariant neurons did not result from biased sampling, we subsampled piriform cortex to match the numbers of olfactory bulb cells. Furthermore, we tested whether relaxing the significance criterion from p<0.01 to p<0.05 would change our results. We find that the fraction of concentration-invariant cells in piriform cortex is consistently and significantly higher than that observed in the olfactory bulb, independent of sampling size and the significance criterion used in our model (*Figure 7—figure supplement 1*, Materials and methods). Note also that, as a consequence of the lower dynamic range of GCaMP3 compared to GCaMP6s (*Chen et al., 2013*), we are likely to overestimate

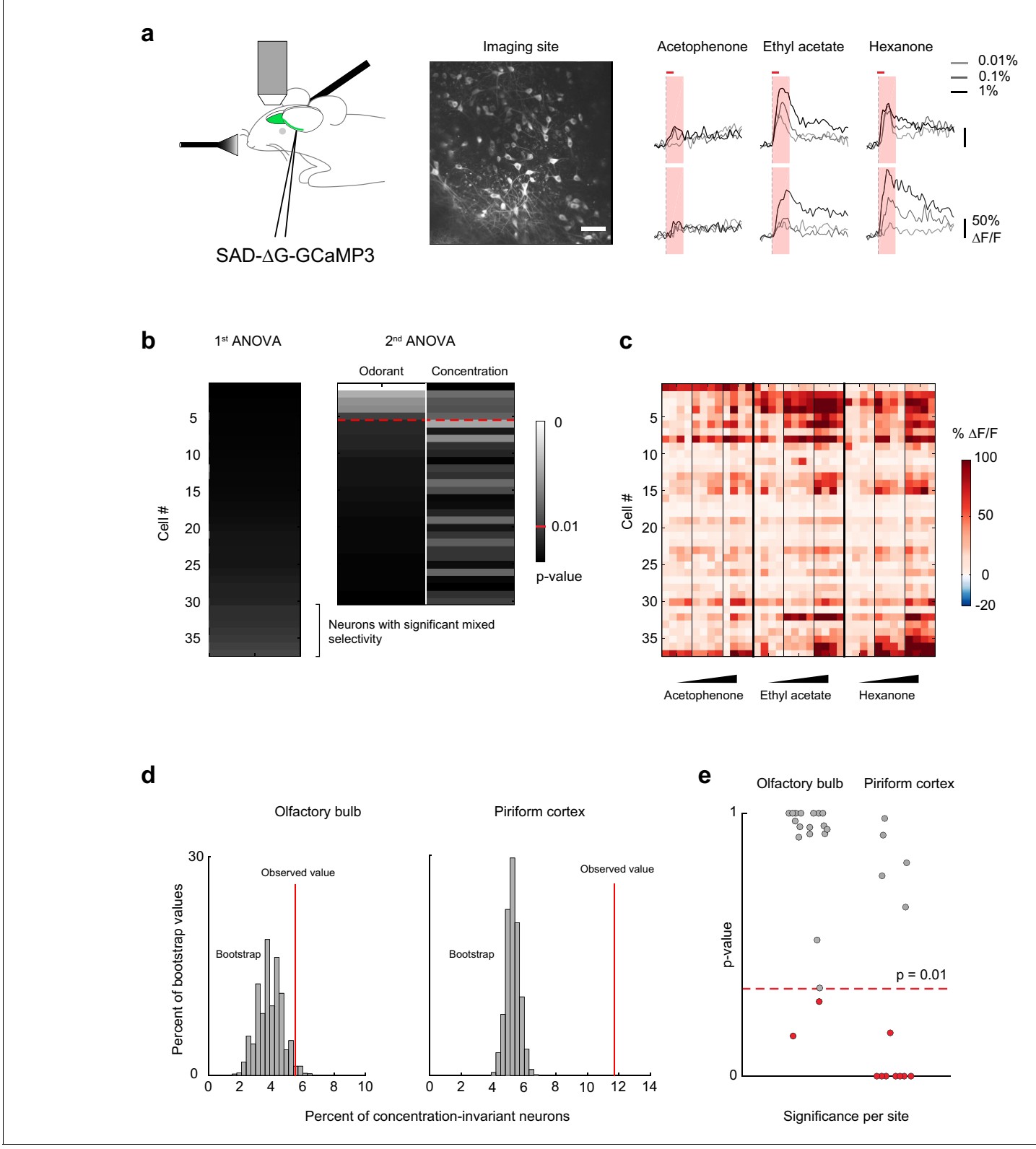

**Figure 7.** Concentration-invariant representations of odor identity emerge in the piriform cortex. (a) (left) Schematic of the experimental protocol. Rabies-GCaMP3 was injected underneath the lateral olfactory tract. After 5–7 days, the olfactory bulb was surgically exposed and mitral cell activity in response to odorants was recorded with two-photon imaging. (middle) Stack average of an imaging site. Scale bar: 50 μm. (right) Deconvolved response traces of two neurons to acetophenone, ethyl acetate and hexanone at three different concentrations (light grey: 1:10,000, dark grey: 1:1,000, black 1:100 vol./vol. dilutions). Red bar: odor presentation. Shaded area: time interval used to integrate ΔF/F values for the analysis of variance in (b). (b)

*Figure 7 continued on next page*

*Figure 7 continued*

Summary of the analysis of variance of the response profiles of a mitral and tufted cell imaging site. See also *Figure 5a*, and Materials and methods for details. (c) Population response of the imaging site in (a) to acetophenone, ethyl acetate and hexanone at three different concentrations (1:10,000, 1:1,000, 1:100 vol./vol. dilutions). Cells are sorted by the p-value of the effect of odorant identity. (d) Percent of concentration-invariant neurons (red line) identified in the olfactory bulb (left, n = 19 imaging sites in eight mice) and in piriform cortex (right, n = 13 imaging sites in 11 mice), overlaid onto the distribution of the percent of concentration-invariant neurons found in the bootstrap samples. (e) p-Values for the number of concentration-invariant neurons identified at each imaging site in the olfactory bulb (left, n = 19) and the piriform cortex (right, n = 13). The number of concentration-invariant neurons is significantly above chance (red dots, p<0.01) in 8 out of 13 imaging sites in the perform cortex, but only in 2 out of 19 imaging sites in the olfactory bulb.

The following figure supplement is available for figure 7:

**Figure supplement 1.** Comparison of the fraction of concentration-invariant neurons in piriform cortex and olfactory bulb.

the concentration-invariance of mitral cell responses, so that the difference in the fraction of concentration-invariant cells in piriform cortex compared to olfactory bulb may be greater than this analysis suggests.

Responses in both piriform and mitral cells are highly heterogeneous. We therefore used a bootstrapping analysis to ensure that the population of concentration-invariant neurons, we observed are indeed over-represented and are not merely a consequence of the inherent response variability across population within each dataset. We repeated the analysis of variance on 10,000 shuffled cell-odor pairs, in which cell identities were scrambled across stimuli but the population statistics for each odorant response were preserved (see Materials and methods). Shuffling mitral cell identities across stimuli indeed resulted in the identification of the same percentage of concentration-invariant neurons observed experimentally (bootstrap mean 3.9 ± 0.8%; observed value 5.5%, p=0.073). By contrast, shuffling cell identities from the piriform dataset yielded significantly fewer concentration-invariant neurons than observed experimentally (bootstrap mean 5.3 ± 0.41%; observed value 11.7%, p<0.001, *Figure 7e*). Finally, we computed the significance of these findings for each individual imaging site. We considered the observed number of concentration-invariant neurons per imaging site to be significantly above chance if it was higher than 99% of the values calculated from the bootstrap samples. We found that the observed number of concentration-invariant neurons in piriform was above chance in 8 out of 13 imaging sites. In contrast, only 2 out of 19 mitral and tufted cell imaging sites contained concentration-invariant neurons above chance level (*Figure 7f*). Together, these results indicate that the encoding of odor identity independent of concentration is robust in piriform cortex, but not in olfactory bulb mitral and tufted cells.

## Discussion

We have examined how the identity of an odor is represented in neural ensemble activity in the piriform cortex using two-photon calcium imaging in anesthetized mice. We found that despite substantial overlap between response patterns evoked by different odorants, odor identity could correctly be predicted from local ensembles of piriform neurons. Furthermore, we observed that piriform response patterns across the population change substantially with increasing odorant concentration, potentially confounding odor identification. However, a substantial fraction of odor-selective piriform neurons exhibit largely concentration-invariant odor responses, and odor identity - independent of intensity - could accurately be decoded from this subpopulation of piriform neurons. Concentration-invariant neurons are overrepresented in piriform cortex but not in olfactory bulb mitral and tufted cells, suggesting that concentration-invariant subnetworks for odor identity emerge in cortical neural circuits for olfaction.

### The structure of odor identity-encoding piriform ensembles

Previous immunohistochemical, electrophysiological and imaging experiments have revealed that piriform odor representations are distributed across a large area of the cortex, and that piriform neurons display discontinuous receptive fields (*Poo and Isaacson, 2009*; *Rennaker et al., 2007*; *Stettler and Axel, 2009*; *Yoshida and Mori, 2007*). These observations led to the speculation that

information about odor is encoded by ensembles of coordinately active neurons distributed across piriform cortex without topographic organization (*Stettler and Axel, 2009*). Here, we explicitly tested this prediction, and we found that odor identity could indeed be decoded from spatially distributed patterns of odor-evoked piriform activity. Our results are consistent with recent data from extracellular recordings in awake rats (*Miura et al., 2012*) and mice (see co-submitted manuscript by Bolding and Franks), which show that odor identity can be accurately decoded from the firing rates of piriform neural ensembles. However, extracellular recordings cannot reveal the spatial organization of the neurons that participate in odor coding. Our data show that odorant-selective neurons do not cluster in space. Furthermore, odor identity can be decoded with similar accuracy from multiple different imaging sites, and information about odor identity appears to be homogeneously distributed within an individual imaging site. Thus, our results provide robust evidence that odor information is encoded without topographic organization by ensembles of piriform neurons.

## Comparison with extracellular recordings in awake mice

*Bolding and Franks (2017)* have used extracellular recordings to explore how odor identity and intensity is encoded in piriform cortex. It is interesting to compare their results with our results obtained from optical imaging. The majority of piriform neurons are narrowly tuned to odor, exhibit high trial-to-trial variability to repeated presentations of the same odor, and substantial overlap between response patterns elicited by different odors. Piriform odor representations are less sparse than previously suggested, and odor identity can accurately be decoded from piriform ensembles in the absence of precise temporal information, consistent with an earlier report (*Miura et al., 2012*). Both studies report that piriform odor representations are not concentration-invariant: odorants presented at a range of different intensities (30-fold concentration range in Bolding and Franks, 100-fold in this study) elicit highly dissimilar response patterns, as dissimilar as the response patterns observed for two different odorants. An important difference between the two studies is that, due to the surgical preparation required to expose piriform cortex for two-photon imaging, optical recordings were performed in ketamine/xylazine-anesthetized mice while electrophysiological recordings were obtained from awake, head-fixed mice. Thus, despite the potentially diverse effects of anesthesia on odor sampling and neural physiology, key features of piriform odor responses identified in the two studies are very similar.

Bolding and Franks utilize the high temporal resolution of electrophysiological recordings to propose that information about odor intensity can be encoded by the synchrony of piriform odor responses. Our study, on the other hand, describes the spatial organization of piriform odor representations and provides evidence for a concentration-invariant subnetwork of piriform neurons that encodes odor identity - independent of odor intensity. Thus, the two studies propose complementary coding schemes for non-interfering representations of odor identity and odor intensity in the mouse olfactory cortex. Interestingly, Bolding and Franks also observe that a subpopulation of neurons exhibits short-latency concentration-invariant odor responses. While differences in the experimental design of the two studies preclude a direct comparison, it is tempting to speculate that these represent the same neural subpopulations.

Anesthesia interferes with active sniffing, and awake mice and rats typically exhibit faster sniff rates that can be dynamically modulated by odor (*Blauvelt et al., 2013*; *Carey and Wachowiak, 2011*; *Wachowiak et al., 2013*). In contrast, we do not observe significant modulation of sniff rate in anesthetized mice. Anesthesia has also been shown to modulate the activity of neurons in the olfactory bulb. In awake mice, periglomerular and granule cell inhibitory neurons exhibit higher levels of activity (*Cazakoff et al., 2014*; *Kato et al., 2012*; *Wachowiak et al., 2013*), while some mitral cells exhibit diminished spontaneous and odor-evoked activity (*Kato et al., 2012*; *Kollo et al., 2014*; *Shusterman et al., 2011*). Anesthesia is likely to differentially affect different types of piriform neurons and has recently been shown to modulate baseline neural activity in piriform cortex (*Tantirigama et al., 2017*). Of note, at elevated odorant concentrations, recordings in awake mice show that the fraction of activated neurons remains stable, while imaging experiments in anesthetized mice show a moderate increase (this study, and *Stettler and Axel, 2009*). This observation suggests that normalization of neural activity is incomplete under anesthesia, consistent with recent reports on the activity of cortical feedback projections to the olfactory bulb (*Boyd et al., 2015*; *Otazu et al., 2015*; *Rothermel and Wachowiak, 2014*). The activity of cortical feedback projections has been suggested to contribute to signal normalization and is attenuated under anesthesia.

## Odor identity and intensity

The ability to accurately determine stimulus identity independent of intensity is critical for olfactory perception and behavior (*Cleland et al., 2011*; *Sirotin et al., 2015*; *Stopfer et al., 2003*). Behavioral experiments have shown that rats can be trained to identify, with high accuracy, monomolecular odorants across a greater than 50,000-fold range in concentration (*Homma et al., 2009*). On the other hand, odor concentration-invariance is not absolute. Behavioral experiments in humans and insects have shown that the perceived identity of an odor can change with concentration (*Bhagavan and Smith, 1997*; *Gross-Isseroff and Lancet, 1988*; *Laing et al., 2003*; *Pelz et al., 1997*). While the perceptual boundaries of odor concentration-invariance remain poorly defined, it is clear that odor identity and intensity must be, at least in part, independently represented in the brain. Earlier experiments in insects have suggested that odor identity and intensity information is intermingled in the antennal lobe. Interestingly, however, multidimensional manifolds representing odor identity emerged after non-linear dimensionality reduction of the data, suggesting that down-stream structures could extract concentration-invariant information about odor identity from antennal lobe activity (*Stopfer et al., 2003*). Moreover, experiments in the fish olfactory bulb suggest that temporal multiplexing could be used to independently transmit odor identity and intensity information to higher olfactory centers in the brain (*Friedrich et al., 2004*) (see also companion manuscript by Bolding and Franks).

We propose an alternative, simple solution for representing these two distinct features of an odor stimulus in the mammalian olfactory cortex; that information about odor identity and odor intensity can be separately represented in distinct subpopulations of piriform neurons. We observed that piriform odor representations change systematically with increasing odorant concentrations, such that responses evoked by an odorant at different concentrations can become as dissimilar as responses evoked by two different odorants. However, we found that ~30% of odor-selective neurons support concentration-invariant odor representations. Such subnetworks can provide a stable representation of odor identity, while information about odor concentration can simultaneously be encoded in other neural ensembles.

What cellular and circuit mechanisms could underlie the generation of concentration-invariant piriform odor responses? The most parsimonious model for such functionally distinct subpopulations of piriform neurons is that these neural subpopulations represent distinct piriform neural cell types. Recent work in acute slice preparations has indeed highlighted the functional diversity of piriform layer II neurons. Piriform layer II cells can be classified into semilunar and superficial pyramidal cells, and cells of intermediate phenotype (*Suzuki and Bekkers, 2011*; *Wiegand et al., 2011*). Superficial semilunar cells have higher input resistance and shorter membrane time constants than pyramidal cells in deep piriform layer II, and semilunar cells receive stronger excitatory input from the olfactory bulb, but weaker associational input than pyramidal cells. Such differences in intrinsic properties and synaptic connectivity could underlie some of the heterogeneity in response types we observe. Interestingly, layer II semilunar and pyramidal cells project to distinct piriform target areas (*Chen et al., 2014*; *Diodato et al., 2016*), providing an opportunity to selectively transmit distinct features of the odor stimulus to different targets. For example, layer II pyramidal cells send cortical feedback projections to the olfactory bulb, which have been implicated in signal normalization - a function that primarily relies on information about stimulus intensity but may be largely independent of odor identity (*Boyd et al., 2015*; *Otazu et al., 2015*). On the other hand, semilunar cells projecting to the cortical amygdala, which has been implicated in the encoding of odor valence (*Root et al., 2014*), may transmit information about odor identity, independent of intensity. The identification of feature-selective subnetworks in piriform cortex, and advances in the characterization of piriform neural connectivity will open up new possibilities for examining odor information routing in cortical neural circuits for olfaction.

## Materials and methods

### Surgical preparation for piriform imaging

Adult (6- to 10-week-old) male mice on a mixed genetic background (C57BL/6; 129Sv) were used for experiments. All experiments were performed according to European and French institutional animal care guidelines (protocol number B750512/00615.02). A total of 400–700 nl of AAV-GCaMP6s (AV-

1-PV2824, Penn Vector, Philadephia, PA) were stereotaxically injected at multiple sites into the piriform cortex at 0.5–1 mm posterior to bregma, using manually controlled pressure injection. 10–13 days later, the piriform cortex was surgically exposed, following experimental procedures described in *Stettler and Axel (2009)*. Briefly, mice were anesthetized with ketamine/xylazine (100 mg/kg/10 mg/kg, Sigma Aldrich) and a head-post was glued to the skull. Skin was retracted to expose the masseter muscle, the superficial temporal vein was cauterized, and the zygomatic bone was removed with fine scissors. The upper portion of the lower jawbone was cleared from tendons and cut out. Minor bleeding was stopped with gelatin sponge (Gelfoam). A well was constructed around the surgical site with silicone sealant (WPI, Sarasola, FL). A small craniotomy, typically 1 × 2 mm in size, was then drilled over the piriform, and the thinned bone was removed with fine forceps. After removal of the dura, the silicone well surrounding the craniotomy was filled with artificial cerebral spinal fluid (ACSF; 125 mM NaCl, 5 mM KCl, 10 mM glucose, 10 mM HEPES, 2 mM CaCl2, 2 mM MgSO4) at all times. A small glass coverslip was placed over the craniotomy and sealed in place using 2% agarose. Six imaging sites in three mice (total number of neurons = 1706) were analyzed for the 13 odorant test panel, and at 13 imaging sites in 11 mice (total number of neurons = 2935) for acetophenone, ethyl acetate and hexanone at three different concentrations.

## Surgical preparation for mitral cell imaging

Methods for mitral and tufted cell imaging are described in *Roland et al., 2016*. Briefly, 3 to 3.5 nL of rabies-GCaMP3 virus was slowly pressure injected underneath the LOT. 5–7 days later, mice were anesthetized using ketamine/xylazine and the skull overlying the olfactory bulb was thinned using a dental drill and removed with forceps, and the dura was peeled back using fine forceps. A small circular glass coverslip was placed over the exposed bulb and sealed in place using 2% agarose. Activity at 19 imaging sites in eight mice (total number of cells = 523) was analyzed.

## Functional imaging

A typical piriform imaging experiment lasted between 2 to 3 hr, and a maximum of three different fields of view were imaged per mouse, at a position between 0 to 1.5 mm posterior to Bregma. Body temperature was maintained at 37°C using a feedback-controlled heating pad (FST). Piriform imaging data were acquired on two different microscopes: a Leica SP5 with a 25x Olympus objective, 256 × 256 pixels for a 347 × 347 µm field of view, and a Scientifica Multiphoton VivoScope with a 20x Olympus objective, 256 × 160 pixels for a 357 × 220 µm field of view. A Mai Tai DeepSee laser (Spectra-Physics, Santa Clara, CA) was tuned to 910 nm. Densely packed piriform layer II neurons at a depth of ~250 µM below the pial surface were imaged. 30 s movie sequences were acquired at a frame rate of ~15 Hz. Mitral cell imaging data were acquired on two different microscopes: Ultima, Prairie Technologies with a 16x objective at 2x zoom or Leica SP5 with a 25x Olympus objective. 25 s movie sequences at 256 × 256 pixels were acquired at a frame rate of 2.53 Hz (Ultima) or 2.9 Hz (Leica SP5).

An odor trial lasted 30 s (8 s of pre-stimulus baseline, 2 s of stimulation, 20 s of post-stimulus acquisition). Odors were delivered at a flow rate of 1 L/min with inter-trial intervals of ~35 s. Odor stimuli for a given experiment consisted of one of two odor sets, delivered through a 16-channel olfactometer (Automate Scientific, Berkley, CA): 13 monomolecular odorants (purchased from Sigma Aldrich at the highest purity available), diluted at 1:10,000 vol./vol. in mineral oil (Sigma Aldrich), and a 'concentration series' consisting of acetophenone, ethyl acetate, and hexanone, at 10-fold increasing concentrations (1:100, 1:1000 and 1:10,000 vol./vol. dilutions). Odorants were presented four times each, in pseudo-randomized order to avoid habituation (average inter-stimulus interval for the same odor stimulus = 7 min). A photoionization detector (miniPID 200B, Aurora Scientific, Canada) was used to confirm reliable odor delivery and to verify that odorant concentration scales according to volumetric ratios. No stimulus was presented twice in a row. For the concentration series experiment, odorant identity was changed at every trial, that is different concentrations of the same odorant were never presented in a row to avoid adaptation.

## Data analysis

### Signal extraction

Data analysis was conducted in Matlab. Motion artifacts were first corrected by using a subpixel translational-based discrete Fourier analysis. Regions of interest (ROIs) were then manually drawn for mitral cell data. For piriform data, ROIs were selected using a semi-automated hierarchical clustering algorithm based on pixel covariance over time (see detailed method below), and the weighted pixel gray value average inside each ROI was used to estimate the fluorescence of single cells at each time frame. The raw fluorescence trace was then upsampled to match the highest sampling rate of each set of experiments (i.e. ~15 Hz for piriform datasets, ~3 Hz for mitral cell datasets). When needed, we corrected for piriform neuropil contamination using a published method (*Kerlin et al., 2010*): the neuropil signal $F_{neuropil}$ (t) surrounding each cell was estimated by averaging the signal of all pixels within a 20 µm circular region from the cell center (excluding all other ROIs). The true fluorescence signal of a cell body was estimated as follows: $F_{true}$ (t) = $F_{measured}$ (t) − r x $F_{neuropil}$ (t), with r = 0.5.

For each trial, the change in fluorescence ($\Delta F/F_0$) was calculated as $(F-F_0)/F_0$, where $F_0$ is the median value between seconds 4 and 8 of the pre-stimulus period. We estimated the baseline fluctuation for a given trial as the standard deviation (SD) of $\Delta F/F_0$ during the baseline period. Odor responses were assessed over a 4-s period following odor onset. A ROC analysis (including blank trials consisting of 30 s of recording with no stimulation for the evaluation of the false-positive rate) was used to determine a threshold with the best sensitivity/specificity ratio. A cell was deemed responsive if it reached and remained above threshold (two times the standard deviation of the baseline) for 21 (activation) or 19 (suppression) frames during this response window. These criteria yielded a detection accuracy (ACC) of 0.92 and 0.915, and a true positive rate (TPR) of 0.89 and 0.87, respectively.

To estimate the time course of firing rate, the calcium signal was temporally deconvolved using the following formula: r(t) = f′(t) + f(t) / τ in which f′(t) is the first derivative of f(t) and τ the decay constant set to 2 s for GCaMP6s (as estimated from the decay of the GCAMP6s fluorescent transients), and 0.5 s for GCaMP3. This signal was low-pass filtered using a four-pole Butterworth filter with a cutoff frequency of 2.5 Hz and used for all subsequent analysis, except for example traces in *Figure 1*, for which the raw signal was plotted.

### Automated cell segmentation

We developed an original method for automatically detecting neurons in the recording region, based on activity time courses. As in other existing methods (*Mukamel et al., 2009*; *Pnevmatikakis et al., 2016*), our method uses temporal activity patterns to automatically segment neurons, including the cells undetectable with contrast-based methods because of low fluorescent baseline. But while other methods focus only on the activity time-courses, our method also takes connexity into account, i.e. that pixels belonging to the same neuron are neighboring each other. We perform iterative clustering of all pixels in the image by merging together neighboring pixels or regions whose activities are correlated.

The clustering procedure is schematized in Figure 1—figure supplement 1. At each step, correlations are calculated for each pair of connected pixels in the entire image (with connections only allowed in the horizontal and vertical directions), and the two pixels with the highest correlation are merged together (Figure 1—figure supplement 1a). The averaged signal of the merged region is then computed and connections and correlations with neighbors are updated. This process continues until all pixels have been merged. We observed that the first pixels merged by this procedure belong to neurons (or axons and dendrites), while pixels belonging to diffuse neuropil regions aggregate at later stages, without altering the segmentation of the neurons (Figure 1—figure supplement 1c). Hence, there is a large range of iteration counts for which we obtain a stable number of clusters corresponding to well segmented neurons and a variable number of regions corresponding to neuropil regions (depending on the stage of the clustering). To obtain a first neuronal segmentation, we therefore selected the clustering output obtained at a particular iteration step. This parameter is user selectable, but the final result is weakly sensitive to it. A value of I = 25,000 was typically chosen for these datasets.

We then extract neurons from this first segmentation step by identifying merged pixel ensembles that have typical neuronal size and shapes (Figure 1—figure supplement 1d). To do so, we compute three measures from each aggregated region and retain only regions falling within a particular range:

1. Size of the region defined as $regsize = \sqrt{\text{number of pixels}}$ must be included between a *minsize* and *maxsize* parameters.
2. Region dispersion defined as $\frac{\sqrt{<||x-center||^2>_{x \in region}}}{regsize}$ must be smaller than a threshold *maxdisp* (typically 0.5): This will select round-shaped regions and discard elongated regions.
3. The ratio between average pixel weights (see below for weights calculation) of the pixels located at the border of the region and of all the pixels should be smaller than a threshold *maxborder* (typically 0.9): This will select regions whose pixel contributions decrease near the border of the region, which is typical of neuron regions but not of neuropil fragment regions.

A graphical interface permits to adjust these selection parameters if necessary. The final segmentation is obtained after a visual quality check in which the user has the possibility to add or remove agglomerated regions within a dedicated graphical interface that helps accelerating the procedure (Figure 1—figure supplement 1d). Importantly, the correlations used in the algorithm are calculated on the full duration of the dataset, but to reduce computational costs, the data is temporally binned into bins of 30 s. Also, two additional preprocessing procedures are applied to remove correlations with large spatio-temporal scales:

1. Global slow drifts are removed by high-pass filtering all time courses with a cutoff period of 100 s.
2. The average signal over all pixels is calculated, and its contribution in each individual pixel (i.e. the projection of the pixel time courses onto this average signal) is subtracted.

To compute the 'average' signals of the regions obtained at the end of the first neuronal segmentation step, we assign weights to each individual pixel in order to obtain the best estimate of the signal common to all these pixels. Note that the regular average corresponds to the case in which all these weights are equal to 1/N (N being the number of pixels inside the region), but does not necessarily constitute the best estimate of the common signal, as pixel with low signal level and comparatively high noise contribute as much as pixels with high signal levels. To find the appropriate weights, we proceed as follows:

1. All weights $w_i$ are initialized to 1/N
2. The signal of the region is calculated as:

$$x_{reg} = \sum_i w_i x_i = Xw$$

(where $x_i$ are column vectors of individual pixel signals)
3. Weights are updated as:

$$W = X^+ x_{reg}$$

(where $X^+ = (X^T X)^{-1} X^T$ is the pseudo-inverse of $X$). Steps 2 and 3 are repeated (typically three times) until convergence.

Note that the weights are then visualized on our graphical interface, highlighting the regions of the neurons that contribute with the strongest signals.

## Population vector analysis

To build response vectors, we averaged the deconvolved $\Delta F/F_0$ signal of all cells over the 4 s following odor onset. This provides an estimate of a neuron's response to each odor trial. We obtained a matrix (piriform neurons x odors trials) representing the population response after odor delivery for every trial. To build the cross-correlation matrix of the patterns of activity, we calculated Pearson's correlation coefficients between every pair of such odor trials.

## Lifetime sparseness

Lifetime sparseness was calculated as:

$$Sl = \frac{1 - \frac{\left[\sum_{j=1}^{N} \frac{r_j}{N}\right]^2}{\sum_{j=1}^{N} \frac{r_j^2}{N}}}{1 - \frac{1}{N}}$$

where $r_j$ are the neurons' responses to individual odors and $N$ is the total number of odors. Lifetime sparseness quantifies the specificity of the neurons' odor-evoked responses (0: uniformly distributed across odors, 1: highly selective for one odor).

## Linear classifier

To quantify the information contained within patterns of piriform activity, we used a linear classifier to predict stimulus identity based upon the response patterns to single odor trials. We obtained comparable classification performance using one-vs-all Support Vector Machine (SVM) with a linear kernel on the raw ΔF/F data, or linear discriminant analysis (LDA) on the principal components encompassing 95% of variance in the data. For computational efficiency, LDA was used for the analysis in Figure S2. To build the response vectors, we accumulated $\Delta F/F_0$ signal over a five frame (~333 ms) sliding window (*Figure 2*), or used a mean response by accumulating signal over the 4 s following odor onset (*Figure 5*). These vectors define a multidimensional (1 neuron = 1 dimension) representation of odors and were used to classify single-trial response patterns. To avoid overfitting, we used a leave-one-out cross-validation strategy, whereby the assessed trial is excluded from the calculation of the centroids.

To assess the decoding of odors based on stimulus identity (*Figure 2*), trials were classified as belonging to one of the stimulus group (13 odors, chance level: 1/13 = 7%). To assess the decoding of odor identity across concentration (*Figure 5*), we adapted a previously published protocol (*Stopfer et al., 2003*). For each odor trial tested, only the most distant concentration was kept to train the classifier (when testing trials at 0.1% concentration, only trials at 0.01% concentration were kept). Thus, outcome of this classification was between seven possible groups of stimuli: one group with the tested odorant identity, and 2 odorants x 3 concentrations groups of the other odors. Classification was deemed correct if the trial was assigned to the correct identity group (1 group out of 7 possible, 1/7 = 14% chance level).

## Analysis of variance of single neurons and bootstrap methods

To investigate the source of modulation of individual neural responses, we used a linear regression approach by fitting an analysis of variance (ANOVA) with concentration, odorant identity, and trial number as fixed effects, using type II sum of square. We defined concentration-invariant cells as cells that were only significantly modulated by odorant identity (Test 1, p<0.01); containing information that enables them to identify at least one of the three odorants, but were not significantly modulated by concentration (Test 2, p>0.01) or by interactions between identity and concentration (Test 3, p>0.01). 'Generic neurons' are all other cells. It is important to note that in our dataset, only 39.9% of all recorded piriform neurons were significantly modulated by odorant identity. Thus, the 11.7% of piriform cells that were identified as concentration-invariant represent ~30% of the neurons that contained significant odorant information. To evaluate the accuracy our method, we estimated an upper bound on the false-positive rate (FPR) of the statistical test. The expected FPR for the intersection of the three tests is the product of the FPRs of each test. Modulation by odor identity (Test 1) is assessed an FPR of 1%, while the FPR for the absence of modulation by intensity (Test 2) and by intensity-identity interactions (Test 3) are not precisely known as they correspond to the false negative rates of the associated tests. However, given that the FPRs for tests 2 and 3 are bounded by 1, the FPR for the concentration-invariant cells is less than 1%. This indicates that at least 10.7% are true positive for concentration invariance (subtracting the 1% FPR). Similarly, of the 5.5% of concentration-invariant neurons identified in the olfactory bulb, at least 4.5% are true positives. To test for the stability of the concentration-invariant subpopulation with respect to the statistical threshold, we repeated the analysis with an alpha-value of 0.05 instead of 0.01. We identified 12.8% of concentration-invariant neurons and, importantly, all concentration-invariant cells previously identified with the 0.01 threshold were included in this ensemble. Thus, the detection of concentration invariant cells is only marginally affected by the exact statistical parameters.

In a second type of analysis, we tested the hypothesis (different from the one tested above) that concentration invariance was a property arising from a random coding scheme, in which response magnitude is arbitrarily assigned for each neuron and odor-concentration pair. We generated 1000 bootstrapped datasets in which we randomly shuffled the cell identities for each odor-concentration pair. In other words, if the data set is described as a 3D array A[Cell_list,Odor_Conc_list,Trials_list], one surrogate dataset AS is generated by performing a randomization of 'Cell_list' for each of the nine items of the Odor_Conc_list. Note that individual trials of a given cell-odor-concentration triplet are kept together. We then submitted the surrogate datasets to the statistical analysis described above for detecting concentration-invariant cells. This enabled us to compute the expected distribution of the fraction of concentration-invariant cells for the 'random coding scheme' hypothesis. We found that the fraction of cells observed in cortex is incompatible with this hypothesis (*Figure 6d*, p<0.001), in contrast to the olfactory bulb (*Figure 6d*, p=0.073).

## Statistics

All descriptive statistics in text and figure legends are mean ± SD. The percent of responding neurons to each stimulus was calculated as the average number of active neurons across four trials. To construct the spatial odor maps, as well as to calculate any parameter implying a thresholding of cell activity (e.g. cell tuning), only cells that responded at least twice out of four trials were included.

## Acknowledgements

We thank Ludovic Cacheux for help with imaging experiments and data analysis, Dmitri Bryzgalov and Simon Daste for help with data analysis, Yves Dupraz for his work on the in vivo imaging set-up, and Jérémie Teillon and Philippe Mailly for help with imaging and imaging data pre-processing. We thank Kevin Bolding, Thomas Preat, Andreas Schaefer, German Sumbre and Jonathan Touboul for critical comments of the manuscript. This work was supported by a Marie Curie International Reintegration grant (IRG 276869), and the 'Amorçage de jeunes équipes' program (AJE201106) of the Fondation pour la Recherche Médicale (to AF), an EMBO short term fellowship (ASTF 395–2014) and a postdoctoral fellowship by the LabEx 'MemoLife' (to BR), by grants from the NIDCD (DC009839 and DC015525) to KMF), grants from the Agence Nationale pour la Recherche (ANR 'SENSEMAKER'), the Marie Curie Program (CIG 334581), and The International Human Frontier Science Program Organization (CDA-0064–2015) (to BB).

## Additional information

### Funding

| Funder | Grant reference number | Author |
| --- | --- | --- |
| European Molecular Biology Organization | ASTF 395 - 2014 | Benjamin Roland |
| LabEx Memolife | | Benjamin Roland |
| National Institute on Deafness and Other Communication Disorders | DC009839 | Kevin M Franks |
| National Institute on Deafness and Other Communication Disorders | DC015525 | Kevin M Franks |
| Agence Nationale de la Recherche | SENSEMAKER | Brice Bathellier |
| Human Frontier Science Program | CDA-0064-2015 | Brice Bathellier |
| Marie Curie Program | CIG 334581 | Brice Bathellier |
| Marie Curie International Reintegration Grant | IRG 276869 | Alexander Fleischmann |

Fondation pour la Recherche     AJE201106                    Alexander Fleischmann
Médicale

The funders had no role in study design, data collection and interpretation, or the decision to submit the work for publication.

## Author contributions

BR, Data curation, Formal analysis, Investigation, Methodology, Writing—review and editing; TD, Data curation, Formal analysis, Methodology, Writing—review and editing; KMF, Conceptualization, Supervision, Writing—review and editing; BB, Conceptualization, Data curation, Formal analysis, Methodology, Writing—review and editing; AF, Conceptualization, Data curation, Formal analysis, Supervision, Funding acquisition, Investigation, Methodology, Writing—original draft, Writing—review and editing

## Author ORCIDs

Benjamin Roland, http://orcid.org/0000-0003-3413-1044
Kevin M Franks, http://orcid.org/0000-0002-6386-9518
Alexander Fleischmann, http://orcid.org/0000-0001-7956-9096

## Ethics

Animal experimentation: This study was performed in strict accordance with French National and INSERM animal care and use committee guidelines (#B750512/00615.02). All surgery was performed under ketamine/xylazine anesthesia.

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
