## [Decision Letter]

[Editors’ note: a previous version of this study was rejected after peer review, but the authors submitted for reconsideration. The first decision letter after peer review is shown below.]

Thank you for submitting your work entitled "Odor concentration-invariant subnetworks in the mouse olfactory cortex" for consideration by *eLife*. Your article has been reviewed by 3 peer reviewers, and the evaluation has been overseen by a Reviewing Editor and a Senior Editor. Our decision has been reached after consultation between the reviewers. Based on these discussions and the individual reviews below, we regret to inform you that your work will not be considered for publication in *eLife* at this time. The reviewers felt that the concerns with the paper were too substantial for acceptance, and would not be feasible to address within a short period. They were unsure if the conclusions would still stand after a re-analysis, but did feel that with substantial changes the authors might be able to address the major issues in a resubmission.

A summary of the key points made by the reviewers is below:

This study looks at large-scale coding using optical recordings and concludesthat a subset of piriform neurons respond in a concentration-invariant way,and can be used for identity coding. The dataset is a potentially useful, broadpopulation readout of piriform cortex activity.

1) The reviewers were concerned that the identification, analysis, andrelevance of the concentration-invariant neurons was not well supported.

The authors should be able to devise a model that would more completelyutilize the data.

2) The authors should provide a strong analysis to support theirassertion that the subsample of cells reliably defined odors in aconcentration-invariant way.

3) There should be a more complete analysis of topography of odor responses.

4) This paper doesn't really draw upon the data from the companion paper,and indeed in some ways doesn't seem to match.

There should be a much better discussion of the effects ofanesthesia, which could compare the two datasets.

The analysis of concentration coding in the two datasets could bedone in a comparative manner, with equivalent analyses in the two cases.

5) The authors should provide a better analysis of the dynamics of responses.

*Reviewer #1:*

This study looks at large-scale coding using optical recordings and concludesthat a subset of piriform neurons respond in a concentration-invariant way,and can be used for identity coding. These are anesthetized recordings, andit would be difficult to do these in awake animals given the quite invasivesurgical preparation.

I review this in context of the companion manuscript by Bolding and Franks.

What do we learn from the current paper that the Bolding manuscript does not say?

The main point here is to identify a subset of concentration-invariant neurons. I'm not sure that this isn't there in the Bolding dataset, but the authors don't seem to have looked for it. I'm surprised that the classification accuracy is so low (69%), with a few hundred neurons on each trial. I'm also surprised at how slowly the classification accuracy builds up and declines. Possibly this is an artifact of the anesthesia.

Overall, I get the sense that the power of optical recordings hasn't beenfully utilized in this study, and that the analysis could be extended toextract more information from the already obtained data. On its own, and inthe present form, the main result of a few cells being concentration-invariant seems limited.

1) The key point of the paper is that piriform odor representations change withincreasing concentrations, but a 10% subset of neurons stay consistent andappear to code only for identity. The authors compare this with an analysisof previous data on OB M/T cells, which are mostly concentration-dependent.

While this result is interesting to olfactory physiologists, I was looking fora more detailed analysis, or a mechanistic insight. For example, ifone confines oneself to fewer odors, does the same subset of neurons stayconsistent? Can the authors devise a model that would span the range fromfully concentration-invariant to concentration-dependent for all odors?

2) I don't see any attempt to relate the optical findings to the electricalrecordings in the companion paper. If one were to filter out the different time-resolutions, would the optical findings simply fall out of the electrical ones? Does the fraction of concentration-independent neurons match?

3) The frame rate for these recordings is quite good, 15 FPS. I am surprisedthere is no analysis of the dynamics of responses for individual neurons,or to compare them between the concentration dependent and independent cells.

4) Did the authors record respiration? Given that the animals were anesthetized, it seems likely that at this frame rate it should be possible to examine respiration dependence of the piriform responses.

5) There are a couple of findings that the authors don't really follow up.

For example, the authors briefly mention functional heterogeneity of PFneuron responses, but they don't seem to follow this up. Also, there isthe finding that the number of responsive Piriform neurons is roughlyindependent of odor concentration.

*Reviewer #2:*

In their manuscript, 'Odor concentration-invariant subnetworks in the mouse olfactory cortex', Roland et al., investigate how ensemble activity patterns in the olfactory cortex decode odor identity vs. intensity. The authors employ multiphoton imaging in anaesthetized mice and probe GCaMP6s responses to different odors and concentrations. They find that odor identity can be accurately decoded from the population activity using a linear classifier. Across different concentrations, the population representations of odorants change, thus degrading information about odor identity. As a solution to this issue, the authors propose that a subset of concentration-invariant cells in the olfactory cortex is well suited to decode odor identity in a concentration invariant manner.

The study is topical and the experiments are carefully performed. However, I have several major concerns with the interpretation of the results presented here, which, in my opinion, preclude the publication of the manuscript in the current form in *eLife*.

1) The authors report ~10% concentration-invariant neurons in the PC vs. ~5% in the OB. They perform a shuffling control and suggest that, unlike the OB, the emergence of concentration-invariant cells in the cortex cannot be attributed to chance sampling. Therefore, these neurons represent an emerging feature of the cortex and can solve the identity problem irrespective of changes in concentration.

The fraction of invariant cells is rather small in both the case of cortex and the bulb. The large majority of neurons in the cortex vary substantially in their responses with changes in concentration (Figure 3). Therefore, to this reviewer, asserting that concentration invariance is solved in particular by these neurons is somewhat more anecdotal rather than based on testable evidence. The authors sample 6 times more neurons in the olfactory cortex compared to the bulb (~3,000 vs. 500). Therefore, it is important to determine whether sub-sampling the olfactory cortex data to match the OB recorded cells, and relaxing the significance criterion from p<0.01 to p<0.05 (Figure 6) change the results presented.

The cross-concentration correlations of odor representations given by even the neurons identified as 'concentration-invariant' are relatively low (0.56, 0.60). What is the average correlation between the neuronal representations of the same odor across repeats, for each of three concentrations sampled? For example, in Figure 5, for ethyl acetate and hexanone, the correlations across repeats for the highest concentration appear by visual inspection higher than 0.8. This value is substantially higher than the correlations reported between the low-medium, medium-high or low-high concentrations in the sampled range. These observations put into question the concentration-invariant status of these neurons.

Can the authors provide a testable model on how the concentration invariant cells are preserved as such across a wider concentration range (for example 50,000 fold vs. 100 fold sampled), and also how this information is readout selectively?

In addition, in the companion paper (Bolding and Franks), in awake animals, a different mechanism (at the whole population level) is proposed for decoding odor identity, while concentration is read by changes in response latency. Can the authors comment how these two perspectives relate to one another?

2) Comparison of olfactory cortex and olfactory bulb activity patterns is hard to interpret in anaesthetized mice since recent reports suggest that cortical feedback is strongly affected by anaesthesia (Rothermel et al., 2014, Boyd et al., 2015, Otazu et al. 2015).

3) Figure 3's title is not supported by the data. Both the percentage of active neurons and their identity change across the sampled concentration range (3B, C). At the level of individual cells, there is substantial variability in lifetime sparseness across concentrations (3D), even though the percentage of neurons that show similar lifetime sparseness values is same across concentrations. In my opinion, such observations invalidate the title's claim that the overall levels of piriform cortex activity remain largely stable across changes in concentration.

*Reviewer #3:*

Concentration invariance is known to be a key property of odor identity coding, and the work of Roland et al. is the first (together with a co-submitted article by Bolding and Franks) to study how piriform cortex solves concentration invariance. Furthermore, the authors use a preparation which allows for simultaneous monitoring of large ensembles of spatially-defined neuronal populations, allowing them to examine how concentration-invariance is implemented at the population level.

The authors demonstrate the presence of concentration-invariant neurons, which comprise a subset of piriform neurons. Additionally, a separate sub-population of piriform neurons change their activity as a function of odor concentration. This is an important observation and is worth publishing. However I have a few comments:

1) One of the novel observations is that the concentration-invariant subnetwork lacks topographic organization in piriform cortex. The authors should elaborate more on this point. The case against topographic organization is made in Figure 2—figure supplement 2 by showing similar classification using a local versus random clustering rule. However, this may not rule out that topography exists. (1) It is possible that any single specific topographic mapping may be washed out in the averaging procedure. The authors should perform an additional analysis: artificially simulating topography and performing the exact same analysis to check that this simulated topography actually yields a classification plot that is different from random. (2) Another possible suggestion to examine topography in piriform representations is to examine the spatial distribution of weights in the classifier.

2) The authors made a point about the trial-to-trial variability of piriform neurons (e.g. Figure 1). A pertinent question is, what is the trial-to-trial variability for neurons in the concentration invariant subnetwork?

3) The author should discuss the effect of anesthesia in their preparations. They compared their research with electrophysiological study co-submitted by Bolding and Franks. The current study revealed the spatial distributions of identity encoding neurons. The authors should discuss the limitations of their approach and clarify what other conclusions can be safely extrapolated for awake case?

[Editors’ note: what now follows is the decision letter after the authors submitted for further consideration.]

Thank you for submitting your article "Odor concentration-invariant subnetworks in the mouse olfactory cortex" for consideration by *eLife*. Your article has been favorably evaluated by a Senior Editor and three reviewers, one of whom is a member of our Board of Reviewing Editors. The reviewers have opted to remain anonymous.

The reviewers have discussed the reviews with one another and the Reviewing Editor has drafted this decision to help you prepare a revised submission.

Summary:

The reviewers felt that the paper was stronger in this resubmission, and that several of the earlier comments had been addressed. Nevertheless, some key points remain.

Essential revisions:

1) The authors must clarify their methods specifically relating to non-randomness of the concentration-invariant responses.

2) The number of cells classified as concentration-invariant seems small and there was some discomfort on whether this population was statistically robust.

3) There must be clarification on what the 'average correlation' of 0.65 refers to: for each odor, or each concentration?

*Reviewer #1:*

In this revised paper, Roland et al. carry out optical recordings from anterior piriform cortex. They have performed substantial additional analyses of this rich dataset. In doing this they have mostly addressed my concerns from the earlier version of the paper.

The point about subsets of neurons involved in concentration-independent coding has been taken up with further analysis and discussion. The analysis on this is much more thorough.

The current version more completely discusses how the optical findings relate to the electrical recordings by Bolding et al.

The authors don't examine respiration dependence of responses, but do explain that the slow dynamics of GCaMP6s would have made this less useful.

The authors also carry out other useful analyses, including spatial organization of responses, and dynamics of responses.

Overall the study is a valuable characterization of population-wide odor responses in the piriform cortex and reveals interesting features of coding.

*Reviewer #2:*

In the second submission of the manuscript "Odor concentration-invariant subnetworks in the mouse olfactory cortex", the authors addressed all reviewers' requests and significantly improved the manuscript. I have only one concern.

In the original review, I requested an independent analysis of different features of the of concertation-invariant and generic cell responses in the piriform cortex. The authors presented results on Figure 7—figure supplement 1 Figure. I was hoping that it will reveal some differences between these two classes of cells. However, variability and temporal profiles of concentration-invariant and generic cell responses were undisguisable. This fact raised a concern about statistical significance of the phenomenon. The authors discussed "non-randomness" of concentration invariant responses. This is a very crucial piece of analysis, and the authors should present a better description for their methods, and some discussion of the results and their limitations. For example, if the observed proportion of concertation invariant responses is 11.7% and proportion of the responses due to random sampling is 5.3%, does it mean that the real concentration-invariant pool of cell is significantly smaller than 11.7%? Or what proportion of concentration invariant responses can be missed by chance, etc.?

*Reviewer #3:*

The authors did perform additional analysis as suggested, but in my opinion, they did not address the main points of criticism. Therefore, I still have concerns regarding the results presented here.

1) The new analysis unfortunately does not fully clarify the points raised related to the concentration invariance. It does show that this group of cells is less concentration variant compared to the entire ensemble of cells imaged. Yet, In the revised version of the manuscript, visual inspection still strongly suggests (Figure 6) that for a given odor, the correlations across repeats for the same (highest) concentration are higher than correlations across repeats for lower concentrations.

In addition, the correlations across repeats for the same (highest) concentration for a given odor are higher than for the same odor across concentrations. It is unclear whether the average correlation (0.65 +/- 0.10) value given across repeats refers indeed to all odors and all concentrations. Is it the same (0.65 +/- 0.10) for each odor and for each concentration? Plotting side by side, for each odor the correlations across concentration (lowest-highest, lowest-intermediate, intermediate-highest), and respectively the average correlations across repeats for each odor concentration (at low, intermediate and highest concentration) would allow a direct comparison for each stimulus.

2) The small fraction of cells that are classified as concentration invariant, remains a concern, as well as lack of evidence that these cells in the data set indeed constitute a cell type as defined by layer, specific input/output patterns or any other features, besides the difference in observed responses (though proposed as such in the manuscript). It is also unclear what the decoding scheme is for a wider concentration range, except an actual change in perceived odor identity mentioned in the text.

It is indeed important to document that a higher percentage of neurons are called as concentration invariant in the piriform cortex compared to the bulb. However, I'm not convinced that the strong message and title of the manuscript should be focused on a result that summarizes, in the best case scenario, the behavior of ~10% of the responsive neurons in the absence of any additional evidence that these 10% of responsive neurons are doing the job as proposed.

3) The anesthetized vs. awake explanation is not robust. In my opinion, different cells can be differentially affected by the brain state, depending on the local and long range inputs they receive and the strength of corresponding activity patterns.

---

## [Author Response]

[Editors’ note: the author responses to the first round of peer review follow.]

*[…] A summary of the key points made by the reviewers is below:*

*This study looks at large-scale coding using optical recordings and concludesthat a subset of piriform neurons respond in a concentration-invariant way,and can be used for identity coding. The dataset is a potentially useful, broadpopulation readout of piriform cortex activity.*

We thank the reviewers for their thoughtful comments and constructive suggestions. The reviewers raise several critical points, which we address by performing extensive additional analyses. These additional tests strongly support and extend our initial conclusions and now provide a much more comprehensive description of our data set.

1) The reviewers were concerned that the identification, analysis, andrelevance of the concentration-invariant neurons was not well supported.

*The authors should be able to devise a model that would more completelyutilize the data.*

We have performed comprehensive additional tests for the selection of concentration-invariant piriform odor responses. We have substantially extended the analysis of our model and discuss how it can accommodate a larger stimulus range. This new analysis provides strong additional support for the relevance of concentration-invariant subnetworks in piriform cortex.

*2) The authors should provide a strong analysis to support theirassertion that the subsample of cells reliably defined odors in aconcentration-invariant way.*

We have addressed this concern directly. We now provide additional analysis of the response properties of concentration-invariant piriform neurons, as well as important additional tests to further demonstrate their ability to encode odor identity – independent of odor intensity.

*3) There should be a more complete analysis of topography of odor responses.*

We have extended our analysis of the topography of piriform odor responses, as suggested by the reviewers. This additional analysis strengthens our initial conclusions.

*4) This paper doesn't really draw upon the data from the companion paper,and indeed in some ways doesn't seem to match.*

*There should be a much better discussion of the effects ofanesthesia, which could compare the two datasets.*

*The analysis of concentration coding in the two datasets could bedone in a comparative manner, with equivalent analyses in the two cases.*

We have added a detailed discussion of the potential effects of anesthesia on piriform odor representations, and we provide additional analysis of concentration coding. Furthermore, we now include an extensive comparison of the two studies and discuss the complementarity of their findings.

*5) The authors should provide a better analysis of the dynamics of responses.*

We have added a detailed analysis of the dynamics of piriform odor responses in the revised version of the manuscript.

Reviewer #1:

*This study looks at large-scale coding using optical recordings and concludesthat a subset of piriform neurons respond in a concentration-invariant way,and can be used for identity coding. These are anesthetized recordings, andit would be difficult to do these in awake animals given the quite invasivesurgical preparation.*

I review this in context of the companion manuscript by Bolding and Franks.

*What do we learn from the current paper that the Bolding manuscript does not say?*

*The main point here is to identify a subset of concentration-invariant neurons. I'm not sure that this isn't there in the Bolding dataset, but the authors don't seem to have looked for it. I'm surprised that the classification accuracy is so low (69%), with a few hundred neurons on each trial. I'm also surprised at how slowly the classification accuracy builds up and declines. Possibly this is an artifact of the anesthesia.*

*Overall, I get the sense that the power of optical recordings hasn't beenfully utilized in this study, and that the analysis could be extended toextract more information from the already obtained data. On its own, and inthe present form, the main result of a few cells being concentration-invariant seems limited.*

According to the reviewer’s suggestion we have substantially extended our analysis and Discussion to better bring into focus the novelty and relevance of our findings, and the complementarity of the optical and electrophysiological approaches.

Our study documents, for the first time, the odor coding properties of piriform neurons based on two- photon calcium imaging data. One driving force behind the development of a comprehensive population coding analysis approach was the observation that piriform odor responses exhibited high levels of heterogeneity and trial-to-trial variability. This key characteristic of piriform odor responses had not been recognized in an earlier piriform imaging paper by Stettler and Axel (Neuron 2009), due to technical limitations and the consequent need to average.

As the reviewer points out, an important finding of our study is the identification of functionally distinct piriform neurons with different odor coding properties. The characterization of distinct coding properties in piriform neurons is a major advance over previous studies, which had treated piriform neurons as homogeneous. Furthermore, our study directly addresses a major unresolved and controversial question in the field. Stettler and Axel provocatively stated that odor information is encoded in randomly distributed ensembles of piriform neurons. This claim has remained untested. We have explicitly tested this prediction, and we now provide new additional analysis to support our finding that odor identity can indeed be decoded from distributed piriform response patterns without spatial clustering. Finally, our imaging data analysis required the development of a novel algorithm for automated cell segmentation of GCaMP6-expressing neurons, which we describe here for the first time. While other methods for cell segmentation have been published recently, the lack of a reliable and efficient means to identify and sort densely packed neurons – in piriform and many other brain areas – has remained a major obstacle in the field. We thus believe that, with the critical new analysis we provide in the revised version of this manuscript, the conceptual and technological advances described in this work and the complementarity of our results in comparison with the results obtained from electrophysiological recordings will of broad interest.

1) The key point of the paper is that piriform odor representations change withincreasing concentrations, but a 10% subset of neurons stay consistent andappear to code only for identity. The authors compare this with an analysisof previous data on OB M/T cells, which are mostly concentration-dependent.

*While this result is interesting to olfactory physiologists, I was looking fora more detailed analysis, or a mechanistic insight. For example, ifone confines oneself to fewer odors, does the same subset of neurons stayconsistent? Can the authors devise a model that would span the range fromfully concentration-invariant to concentration-dependent for all odors?*

We thank the reviewer for this interesting idea. According to the reviewer’s suggestion, we have extended our analysis of concentration invariance. First, we identified concentration-invariant neurons across a 10-fold instead of a 100-fold range in odorant concentration. Using the same selection criteria as for the original data set, this analysis yielded 19.2% ( ± 7.4% SD across experiments) concentration-invariant neurons. Second, we have identified concentration-invariant neurons for pairs of two odorants instead of the three odorants in our test panel. We found that 22% ( ± 8% SD) of the cells were concentration-invariant for at least one pair of odorants. Of those cells, 49% were identified as concentration-invariant for all three odorants while the response of the other 51% of cells was modulated by the concentration of one of the three odorants. We have included this additional analysis in the revised manuscript (Results, subsection “A concentration-invariant piriform subnetwork”, fourth paragraph).

These data suggest that concentration-invariant subnetworks of neurons remain relatively stable within the range of our stimulus set and can provide concentration-invariant odor identity information across a large range of concentration, but that concentration-invariance is not absolute. Concentration-invariant subnetworks depend, to a certain extent, on stimulus complexity and concentration range. This model is consistent with behavioral experiments in humans and insects, which have shown that the perceived odor identity can vary with concentration and stimulus complexity (Bhagavan and Smith, 1997; Engen, 1964; Gross-Isseroff and Lancet, 1988; Laing et al., 2003). We have included a more extensive discussion of odor concentration invariance in the revised manuscript (Discussion, subsection “Odor identity and intensity”, first paragraph). In future experiments, it will be interesting to establish behavioral tests combined with electrophysiological or optical recordings to quantitatively define the boundaries of concentration-invariance. However, these experiments are beyond the scope of this manuscript.

*2) I don't see any attempt to relate the optical findings to the electricalrecordings in the companion paper. If one were to filter out the different time-resolutions, would the optical findings simply fall out of the electrical ones? Does the fraction of concentration-independent neurons match?*

We thank the reviewer for his/her suggestion to more directly relate our findings to the results obtained from extracellular recordings by Bolding and Franks. We have added additional analyses, and we carefully discuss the similarities and differences of the two studies and their complementarity (Discussion, subsection “Comparison with extracellular recordings in awake mice”).

Bolding and Franks utilize the high temporal resolution of their electrophysiological recordings to propose that information about odor intensity is encoded in the synchrony of piriform odor responses. Our study, on the other hand, focuses on the spatial organization of piriform odor representations and provides evidence for a concentration-invariant subpopulation of piriform neurons, which encodes odor identity – independent of odor intensity. Thus, the two studies propose complementary coding schemes for non-interfering representations of odor identity and odor intensity in the mouse olfactory cortex.

Interestingly, Bolding and Franks report that a subpopulation of ~20% of neurons exhibits concentration-invariant odor response latencies. Differences in the experimental design of the two studies, including the number and concentration range of odorants and the potential bias of extracellular recordings for active neurons preclude a direct comparison of the fraction of concentration-invariant neurons. However, the results of the two studies are compatible and it is tempting to speculate that they indeed represent the same type of piriform neuron.

A difference between the two studies is anesthesia: due to the surgical preparation required to expose piriform cortex for two-photon imaging, optical recordings were performed in ketamine/xylazine-anesthetized mice while electrophysiological recordings were obtained from awake, head-fixed mice. Despite the diverse effects of anesthesia on odor sampling and olfactory processing, many key parameters of piriform odor responses are very similar. The majority of piriform neurons are narrowly tuned to odor, exhibit high trial-to-trial variability to repeated presentations of the same odor, and substantial overlap between response patterns elicited by different odors. Piriform odor representations are less sparse than previously suggested, and odor identity can accurately be decoded from piriform ensembles in the absence of precise temporal information, consistent with a previous report by the Uchida lab (Miura et al., 2012). Importantly, both studies then report that piriform odor representations are not concentration-invariant: odorants presented at a range of different intensities (30-fold concentration range in Bolding and Franks, 100-fold in this study) elicit highly dissimilar response patterns, as dissimilar as the response patterns observed for two different odorants.

*3) The frame rate for these recordings is quite good, 15 FPS. I am surprisedthere is no analysis of the dynamics of responses for individual neurons,or to compare them between the concentration dependent and independent cells.*

We thank the reviewer for his/her suggestion to explore the dynamics of piriform odor responses. We now include a new analysis of population dynamics over time (Figure 2). We trained a classifier at defined time points after odor onset, and we measured classification accuracy over time. This analysis shows that odor representations are dynamically rearranged to maintain odor information for several seconds after odor offset.

According to the reviewer’s suggestion, we also provide additional analyses on the dynamics of piriform odor responses. We have characterized response onset, rise time and duration, for concentration-invariant and concentration-dependent cells (Figure 6—figure supplement 1). Our analysis shows that concentration-invariant neurons exhibit response profiles similar to those of other odor-responsive neurons. However, it is important to note that the response dynamics of GCaMP6s (rise time to peak: ~0.5 s, half decay time: >1 s; Chen at el., Nature 2013) are likely to be the limiting factor in our analysis of response dynamics analysis, rather than the frame rate of our recordings.

*4) Did the authors record respiration? Given that the animals were anesthetized, it seems likely that at this frame rate it should be possible to examine respiration dependence of the piriform responses.*

We obtained reliable recordings of respiration frequency in the majority of our experiments (11 of 14 mice, 15 of 19 imaging sites). We observe an average breath period of 370 ms (Figure 4—figure supplement 1). This breath frequency is very similar to that observed by Bolding and Franks in awake mice. However, while Bolding and Franks observe a small increase of breath frequency at increasing odor concentrations, breath frequency in anesthetized mice did not change following odor delivery at any concentration (breath period, mean ± SD: 0.01%: 0.36 ± 0.05, 0.1%: 0.37 ± 0.06, 1%:

0.37 ± 0.05). We have included this additional information in Figure 4—figure supplement 1 of the revised manuscript.

We also provide additional analysis of population dynamics over time (Figure 2). However, due to the slow temporal dynamics of the GCaMP6s calcium indicator, and given that respiration in our experiments is independent from odor exposure, we did not examine respiration dependence of piriform odor representations.

5) There are a couple of findings that the authors don't really follow up.

*For example, the authors briefly mention functional heterogeneity of PFneuron responses, but they don't seem to follow this up. Also, there isthe finding that the number of responsive Piriform neurons is roughlyindependent of odor concentration.*

According to the reviewer’s suggestion, we have added a detailed characterization of the response properties of piriform neurons (temporal dynamics, change in fluorescence, trial-to-trial variability), which further document their functional heterogeneity. We now include additional analysis to describe the concentration-dependence of piriform odor representations and its implications for odor coding in the revised manuscript. Please note that while the number of responsive piriform neurons only moderately changes with odor concentration, the individual response properties of the majority of neurons are odor concentration-dependent. We discuss this in the manuscript.

Reviewer #2:

*In their manuscript, 'Odor concentration-invariant subnetworks in the mouse olfactory cortex', Roland et al., investigate how ensemble activity patterns in the olfactory cortex decode odor identity vs. intensity. The authors employ multiphoton imaging in anaesthetized mice and probe GCaMP6s responses to different odors and concentrations. They find that odor identity can be accurately decoded from the population activity using a linear classifier. Across different concentrations, the population representations of odorants change, thus degrading information about odor identity. As a solution to this issue, the authors propose that a subset of concentration-invariant cells in the olfactory cortex is well suited to decode odor identity in a concentration invariant manner.*

*The study is topical and the experiments are carefully performed. However, I have several major concerns with the interpretation of the results presented here, which, in my opinion, preclude the publication of the manuscript in the current form in eLife.*

*1) The authors report ~10% concentration-invariant neurons in the PC vs. ~5% in the OB. They perform a shuffling control and suggest that, unlike the OB, the emergence of concentration-invariant cells in the cortex cannot be attributed to chance sampling. Therefore, these neurons represent an emerging feature of the cortex and can solve the identity problem irrespective of changes in concentration.*

*The fraction of invariant cells is rather small in both the case of cortex and the bulb. The large majority of neurons in the cortex vary substantially in their responses with changes in concentration (Figure 3). Therefore, to this reviewer, asserting that concentration invariance is solved in particular by these neurons is somewhat more anecdotal rather than based on testable evidence. The authors sample 6 times more neurons in the olfactory cortex compared to the bulb (~3,000 vs. 500). Therefore, it is important to determine whether sub-sampling the olfactory cortex data to match the OB recorded cells, and relaxing the significance criterion from p<0.01 to p<0.05 (Figure 6) change the results presented.*

*The cross-concentration correlations of odor representations given by even the neurons identified as 'concentration-invariant' are relatively low (0.56, 0.60). What is the average correlation between the neuronal representations of the same odor across repeats, for each of three concentrations sampled? For example, in Figure 5, for ethyl acetate and hexanone, the correlations across repeats for the highest concentration appear by visual inspection higher than 0.8. This value is substantially higher than the correlations reported between the low-medium, medium-high or low-high concentrations in the sampled range. These observations put into question the concentration-invariant status of these neurons.*

*Can the authors provide a testable model on how the concentration invariant cells are preserved as such across a wider concentration range (for example 50,000 fold vs. 100 fold sampled), and also how this information is readout selectively?*

*In addition, in the companion paper (Bolding and Franks), in awake animals, a different mechanism (at the whole population level) is proposed for decoding odor identity, while concentration is read by changes in response latency. Can the authors comment how these two perspectives relate to one another?*

We thank the reviewer for bringing up several important points.

First, the reviewer suggests crucial additional tests to strengthen our assertion that concentration- invariant neurons are overrepresented in piriform cortex compared to olfactory bulb. According to the reviewer’s suggestion, we subsampled piriform cortex to balance the numbers of neurons. Furthermore, we tested our model with a relaxed significance criterion of p < 0.05 (instead of p < 0.01). The results strongly support our initial conclusions. We performed 1000 random iterations of subsampling of piriform neurons and obtained in all cases a fraction of concentration-invariant neurons larger than the one observed for the olfactory bulb. We also performed a bootstrap analysis as in Figure. 7D and found in all subsamplings that the observed fraction of concentration invariant neurons was significantly larger than chance (p<0.01). Furthermore, relaxing the selection criterion to p < 0.05 yields 12.7% ± 5.6% of concentration-invariant piriform neurons, significantly more than in the olfactory bulb (6.7 ± 6.1%, Wilcoxon test, p=0.0076). This additional analysis is now presented in Figure 7—figure supplement 1 of the revised manuscript.

The reviewer points out that correlations of odor representations by concentration-invariant neurons appear lower across concentrations than across individual trials. The average correlations for the three odorants across individual trials for each of the three concentrations are 0.65 ( ± 0.10). The average correlations for the three odorants across the three concentrations are 0.6 ( ± 0.09) (low to intermediate) and 0.56 ( ± 0.1) (intermediate to high) for the concentration-invariant neurons, compared to 0.53 ( ± 0.12) and 0.36 ( ± 0.12) for the entire piriform ensemble. We now include this additional information and a more detailed discussion of concentration-invariance in the revised version of the manuscript (Results: subsection “A concentration-invariant piriform subnetwork”, Discussion: subsection “Odor identity and intensity”).

Our analysis suggests that odor representations comprised of concentration-invariant neurons are significantly more similar across the 100-fold range of odor concentration than the entire piriform ensemble, and that they provide a computational advantage to identify odors independent of concentration (Figure 5). However, the concentration-invariant subnetworks we identify are not fully concentration-invariant across the tested range in odorant concentration (see also reviewer 1, point 1). The ability to identify an odor across a range of concentrations is essential for olfactory- driven behaviors. However, behavioral studies in humans and insects have also shown that some odors can change perceived odor identity with changing odor concentrations (Bhagavan and Smith, Physiol Behav. 1997; Engen, Ann N Y Acad Sci. 1964; Gross-Isseroff and Lancet, Chem. Senses 1988; Laing et al., 2003; Peltz et al., 1997). Thus, odor representations must provide representations of odor identity that are typically robust to changes in odorant concentration, but concentration- invariance is not absolute. It will be interesting to establish behavioral tests combined with electrophysiological or optical recordings to further examine the boundaries of concentration- invariance. However, these experiments are beyond the scope of this manuscript.

Finally, according to the reviewer’s suggestion, we now discuss how concentration-invariant information encoded in piriform neural subnetworks may be read out by downstream brain areas, and how our model complements the findings by Bolding and Franks, who propose a multiplexed spatial and temporal coding strategy for representing distinct odor features. In brief, we propose that functionally distinct subpopulations of piriform neurons represent distinct piriform neural cell types. Recent work in acute slice preparations has indeed highlighted the diversity of piriform layer II cells. Piriform layer II is composed of semilunar and superficial pyramidal cells, and cells of intermediate phenotype (Suzuki and Bekkers, 2011; Wiegand et al., 2011). Differences in intrinsic properties and synaptic connectivity could explain how different piriform neural cell types can represent different stimulus features. Moreover, layer II semilunar and pyramidal cells project to distinct piriform target areas (Chen et al., 2014; Diodato et al., 2016), providing an opportunity to selectively read out stimulus information. For example, layer II pyramidal cells send cortical feedback projections to the olfactory bulb, which have been implicated in signal normalization – a function that primarily relies on information about stimulus intensity but may be largely independent of odor identity (Boyd et al., 2015; Otazu et al., 2015). On the other hand, semilunar cells projecting to the cortical amygdala, which has been implicated in the encoding of odor valence (Root et al., 2014), may transmit information about odor identity, independent of intensity.

*2) Comparison of olfactory cortex and olfactory bulb activity patterns is hard to interpret in anaesthetized mice since recent reports suggest that cortical feedback is strongly affected by anaesthesia (Rothermel et al., 2014, Boyd et al., 2015, Otazu et al. 2015).*

We agree with the reviewer’s point that anesthesia affects the activity cortical feedback projections to the olfactory bulb. An important finding of our study is that odor identity independent of odor intensity can be decoded from a functionally distinct subnetwork of piriform neurons, and that concentration- invariant cells are overrepresented in piriform cortex compared to the olfactory bulb. The most parsimonious model for concentration-invariant subnetworks in piriform cortex – and for functionally distinct subpopulations of piriform neurons more generally – is that such neural subpopulations represent distinct piriform neural cell types. Work in acute slice preparations has indeed highlighted the diversity of intrinsic properties and synaptic connectivity of piriform layer II cells (Choy et al., 2015; Johenning et al., 2009; Large et al., 2016; Suzuki and Bekkers, 2011).

We propose that the functional differences of odor responses we observe reflect the response properties of distinct piriform neural cell types. While anesthesia can modulate neural odor responses, it is unlikely that anesthesia imposes functional differences between different neural cell types. Future experiments will be required to directly test the extent to which our model is valid across brain states.

*3) Figure 3's title is not supported by the data. Both the percentage of active neurons and their identity change across the sampled concentration range (3B, C). At the level of individual cells, there is substantial variability in lifetime sparseness across concentrations (3D), even though the percentage of neurons that show similar lifetime sparseness values is same across concentrations. In my opinion, such observations invalidate the title's claim that the overall levels of piriform cortex activity remain largely stable across changes in concentration.*

We thank the reviewer for pointing out this potential confound. We have changed the figure caption to: Figure 4. Odor-evoked activity and sparseness of individual piriform neurons depends on odorant concentration.

Reviewer #3:

*Concentration invariance is known to be a key property of odor identity coding, and the work of Roland et al. is the first (together with a co-submitted article by Bolding and Franks) to study how piriform cortex solves concentration invariance. Furthermore, the authors use a preparation which allows for simultaneous monitoring of large ensembles of spatially-defined neuronal populations, allowing them to examine how concentration-invariance is implemented at the population level.*

*The authors demonstrate the presence of concentration-invariant neurons, which comprise a subset of piriform neurons. Additionally, a separate sub-population of piriform neurons change their activity as a function of odor concentration. This is an important observation and is worth publishing. However I have a few comments:*

*1) One of the novel observations is that the concentration-invariant subnetwork lacks topographic organization in piriform cortex. The authors should elaborate more on this point. The case against topographic organization is made in Figure 2—figure supplement 2 by showing similar classification using a local versus random clustering rule. However, this may not rule out that topography exists. (1) It is possible that any single specific topographic mapping may be washed out in the averaging procedure. The authors should perform an additional analysis: artificially simulating topography and performing the exact same analysis to check that this simulated topography actually yields a classification plot that is different from random. (2) Another possible suggestion to examine topography in piriform representations is to examine the spatial distribution of weights in the classifier.*

We thank the referee for bringing up this important point. To more quantitatively test the spatial organization of odor-responsive piriform neurons we have developed a statistical test to detect spatial clustering of functionally distinct groups of cells. We first define functionally distinct neurons by their odor tuning, and we compute the nearest neighbor index (NNI) as the mean distance of each cell to its nearest neighbor within the same group. We then test whether the observed NNI is significantly smaller than for neurons whose positions are shuffled to generate the expected NNI distribution for randomly distributed ensembles. Furthermore, we simulate subtle inhomogeneities in the spatial distribution of neurons to validate the sensitivity of this test to detect spatial clustering. This additional analysis is now presented in Figure 3 and fails to detect spatial organization (p > 0.05) in piriform response patterns to 13 different odorants for all 6 imaging sites.

We then use the same model to test the spatial organization of concentration-invariant piriform subnetworks. A random distribution of neurons (p > 0.05) was again observed for 11 out of 13 imaging sites, and only weak clustering was observed for 2 imaging sites. To illustrate these findings we now show results of the statistical tests and the distributions of concentration-invariant neurons for each imaging sites in Figure 6—figure supplement 1.

We favor to keep our analysis of odor classification performance across increasingly large patches of cells as a complementary, more qualitative illustration of the fact that odor identity information is broadly distributed across piriform ensembles. Please note that homogenous classification performance can be obtained even with spatial clustering of odor identity preference, thus we do not suggest this analysis to be a proof of the lack of spatial organization. Finally, we considered the idea of using classifier weights as a test for statistical significance. However, with 13 odorants and one classifier for each odorant pair it is difficult to meaningfully interpret the distribution of the 13 corresponding sets of weights.

*2) The authors made a point about the trial-to-trial variability of piriform neurons (e.g. Figure 1). A pertinent question is, what is the trial-to-trial variability for neurons in the concentration invariant subnetwork?*

According to the reviewer’s suggestion we have determined trial-to-trial variability of concentration- invariant neurons (Figure 6—figure supplement 1). We do not observe a significant difference in trial- to-trial variability between concentration-invariant and ‘generic’ neurons.

*3) The author should discuss the effect of anesthesia in their preparations. They compared their research with electrophysiological study co-submitted by Bolding and Franks. The current study revealed the spatial distributions of identity encoding neurons. The authors should discuss the limitations of their approach and clarify what other conclusions can be safely extrapolated for awake case?*

We thank the reviewer for this suggestion. We have included a detailed discussion of potential anesthesia effects, and a direct comparison with the results obtained from electrophysiological recordings in awake mice by Bolding and Franks in the revised manuscript.

As the reviewer points out, an important difference between the two studies is anesthesia: due to the surgical preparation required to expose piriform cortex for two-photon imaging, optical recordings were performed in ketamine/xylazine-anesthetized mice while electrophysiological recordings were obtained from awake, head-fixed mice. Despite the diverse effects of anesthesia on odor sampling and olfactory processing (see below), many key parameters characteristic of piriform odor responses are very similar. Piriform odor representations are relatively sparse, and the majority of piriform neurons are narrowly tuned to odor. Importantly, piriform odor representations exhibit high trial-to-trial variability to repeated presentations of the same odor and substantial overlap between response patterns elicited by different odors. Finally, and consistent with an earlier report (Miura et al., 2012), odor identity can accurately be decoded from piriform ensembles in the absence of precise temporal information. Both studies then report that piriform odor representations are not concentration-invariant: odorants presented at a range of different intensities (30-fold concentration range in Bolding and Franks, 100-fold in this study) elicit highly dissimilar response patterns, as dissimilar as the response patterns observed for two different odorants.

Bolding and Franks utilize the high temporal resolution of their electrophysiological recordings to propose that information about odor intensity is encoded in the synchrony of piriform odor responses. Our study, on the other hand, focuses on the spatial organization of piriform odor representations and provides evidence for a concentration-invariant subnetwork of piriform neurons that encodes odor identity – independent of odor intensity. Thus, the two studies propose complementary coding schemes for non-interfering representations of odor identity and odor intensity in the mouse olfactory cortex. Interestingly, Bolding and Franks also observe a subpopulation of neurons that exhibit short- latency concentration-invariant odor responses. While differences in the experimental design of the two studies preclude a direct comparison of these neural subpopulations, it is tempting to speculate that they represent the same type of piriform neuron.

Anesthesia interferes with active sniffing, and awake mice and rats generally exhibit faster sniff rates that can be dynamically modulated by odor (Blauvelt et al., 2013; Carey and Wachowiak, 2011; Wachowiak et al., 2013). In contrast, we do not observe significant modulation of sniff rate in ketamine/xylazine-anesthetized mice. Anesthesia has also been shown to modulate the activity of neurons in the olfactory bulb. In awake mice, periglomerular and granule cell inhibitory neurons exhibit higher levels of activity (Cazakoff et al.et al., 2014; Kato et al., 2012; Wachowiak et al., 2013), while some mitral cells exhibit diminished spontaneous and odor-evoked activity (Kato et al., 2012; Kollo et al., 2014; Shusterman et al., 2011). Tantirigama et al. (2017) report that ketamine/xylazine anesthesia affects baseline activity of piriform neurons. In contrast, unpublished data by Bolding and Franks (manuscript in preparation) suggests that ketamine/xylazine anesthesia has no significant effect on the fraction of odor-responsive piriform neurons or their odor tuning (see Figure 8). However, anesthesia does alter sniff coupling and the temporal response properties of piriform neurons (Bolding and Franks, companion paper and Figure 8). Of note, at elevated odorant concentrations, extracellular recordings in awake mice show that the fraction of activated neurons remains stable, while optical recordings in anesthetized mice show a moderate increase. This observation suggests that normalization of neural activity is incomplete under anesthesia, consistent with recent reports on the activity of cortical feedback projections to the olfactory bulb (Boyd et al., 2015; Otazu et al., 2015; Rothermel and Wachowiak, 2014). The activity of cortical feedback projections has been suggested to contribute to signal normalization and is attenuated under anesthesia. We now discuss this distinction in our paper.

Author response image 1.Odor responses are similar in awake and anesthetized mice.(**A**) Example showing respiration and spontaneous activty in a mouse before (black) and during ketamine/xylazine (k/x, cyan) anesthesia. (**B**) Example PSTHs in response to 3 odors before and during anesthesia. Each row is a different cell. Responses are aligned to inhalation. (**C**) Population and lifetime sparseness, averaged across experiments, are similar in awake and during k/x anesthesia. (**D**) Distribution of peak firing rates are similar in awake and anesthetized mice. (**E**) Responses awake mice occur earlier than under anesthesia. (**F**) Response durations are similar in awake and anesthetized mice.**DOI:**
http://dx.doi.org/10.7554/eLife.26337.015

The most parsimonious model for concentration-invariant subnetworks in piriform cortex – and for functionally distinct subpopulations of piriform neurons more generally – is that such neural subpopulations represent distinct piriform neural cell types. Work in acute slice preparations has indeed highlighted the diversity of intrinsic properties and synaptic connectivity of piriform layer II cells (Choy et al., 2015; Johenning et al., 2009; Large et al., 2016; Suzuki and Bekkers, 2011). However, experiments investigating the in vivo odor response properties of piriform neurons have treated these different cell types as homogeneous (Miura et al., 2012; Poo and Isaacson, 2009; Rennaker et al., 2007; Stettler and Axel, 2009), and the response properties of distinct piriform principle cells to odor remain unknown We favor the model that the functional differences we observe reflect the response properties of distinct piriform neural cell types and are thus unlikely to be imposed by anesthesia. Future experiments will be required to directly test the extent to which this model is valid across brain states.

[Editors' note: the author responses to the re-review follow.]

*Essential revisions:*

*1) The authors must clarify their methods specifically relating to non-randomness of the concentration-invariant responses.*

We have added a comprehensive description of the statistical analysis of concentration-invariant neurons to the Methods section of the revised manuscript. We have also included a detailed discussion of the results and limitations of this analysis.

*2) The number of cells classified as concentration-invariant seems small and there was some discomfort on whether this population was statistically robust.*

There are essentially two concerns here: the seemingly small size of the concentration-invariant cells and the robustness of the statistics used to identify them.

Regarding the size of the concentration-invariant subpopulation, it is important to bear in mind that in our data set, ~40% of the cells responded selectively to any of the three odorants. Thus, the ~10% of cells that have concentration-invariant responses represent almost one third of all odor-selective cells. While this was described in the Methods section and further explored in the additional analysis of concentration-invariant subpopulations requested by reviewer 1, it was not explicitly stated in the text. We have now clarified this point in the Results section and Discussion of the revised manuscript.

We have also extended the analysis and discussion of the statistics used for the identification of concentration-invariant piriform neurons. Importantly, cross-correlations, PCA and linear classification analysis consistently support the robustness of our conclusions.

*3) There must be clarification on what the 'average correlation' of 0.65 refers to: for each odor, or each concentration?*

We apologize for confusion here. We now provide a detailed description of the cross-correlation analysis in Figure 6—figure supplement 1. Following reviewer 3’s recommendation, we have plotted, side by side, the cross-correlation values within and across odorant concentrations for each odorant. We see how this helps clarify the issue and thank the reviewer for this suggestion.

*Reviewer #2:*

*In the second submission of the manuscript "Odor concentration-invariant subnetworks in the mouse olfactory cortex", the authors addressed all reviewers' requests and significantly improved the manuscript. I have only one concern.*

*In the original review, I requested an independent analysis of different features of the of concertation-invariant and generic cell responses in the piriform cortex. The authors presented results on Figure 7—figure supplement 1. I was hoping that it will reveal some differences between these two classes of cells. However, variability and temporal profiles of concentration-invariant and generic cell responses were undisguisable. This fact raised a concern about statistical significance of the phenomenon. The authors discussed "non-randomness" of concentration invariant responses. This is a very crucial piece of analysis, and the authors should present a better description for their methods, and some discussion of the results and their limitations. For example, if the observed proportion of concertation invariant responses is 11.7% and proportion of the responses due to random sampling is 5.3%, does it mean that the real concentration-invariant pool of cell is significantly smaller than 11.7%? Or what proportion of concentration invariant responses can be missed by chance, etc.?*

We appreciate the reviewer’s suggestion to provide a comprehensive description of the statistical methods used to characterize concentration-invariant piriform neurons. We have added a detailed explanation of the bootstrap analysis used for the characterization of concentration-invariant subpopulations of piriform neurons in the Methods section of the revised manuscript. We also discuss the results obtained from this analysis and its limitation (subsection “Analysis of variance of single neurons and bootstrap methods”).

*Reviewer #3:*

*The authors did perform additional analysis as suggested, but in my opinion, they did not address the main points of criticism. Therefore, I still have concerns regarding the results presented here.*

*1) The new analysis unfortunately does not fully clarify the points raised related to the concentration invariance. It does show that this group of cells is less concentration variant compared to the entire ensemble of cells imaged. Yet, In the revised version of the manuscript, visual inspection still strongly suggests (Figure 6) that for a given odor, the correlations across repeats for the same (highest) concentration are higher than correlations across repeats for lower concentrations.*

*In addition, the correlations across repeats for the same (highest) concentration for a given odor are higher than for the same odor across concentrations. It is unclear whether the average correlation (0.65 +/- 0.10) value given across repeats refers indeed to all odors and all concentrations. Is it the same (0.65 +/- 0.10) for each odor and for each concentration? Plotting side by side, for each odor the correlations across concentration (lowest-highest, lowest-intermediate, intermediate-highest), and respectively the average correlations across repeats for each odor concentration (at low, intermediate and highest concentration) would allow a direct comparison for each stimulus.*

We thank reviewer 3 for pointing out additional points for clarification.

Regarding the correlation matrices presented in Figure 5 and Figure 6, it is important to keep in mind two principles. First, this presentation of the data is not aimed at assessing the discriminability of representations (this must be done with cross-validated classifiers, as we did in Figure 6) but rather is presented as a way to qualitatively visualize the variability of population responses. Second, as a rule of thumb (for details, see Bathellier et al., Neuron 2012), one can consider the response patterns for two stimuli significantly distinct from each other if their correlations are lower than the trial-to-trial correlations of repeated presentations of either stimulus alone.

As suggested by the reviewer, we have plotted, side by side, the cross-correlations across trials and concentrations for each odorant (Figure 6—figure supplement 1). Indeed, this presentation helps clarify our conclusion that, for all three odorants, responses of concentration-invariant piriform neurons are highly correlated across trials and concentrations.

Despite an apparent trend (which disappears when using concentration-invariant subpopulations identified with p < 0.05 as the significance criterion), there is no statistically significant difference (Wilcoxon ranked sum test p > 0.05, see new Figure 6—figure supplement 1) between trial-to-trial correlations for responses to odorants at low (0.01%) versus high (1%) concentrations. Furthermore, we tested if trial-to-trial correlations at low odorant concentrations were different from the low versus high odorant concentration correlations. We found no significant differences for acetophenone and hexanone, and only a small but significant difference for ethyl acetate (0.57 ± 0.04% SD versus 0.65 ± 0.05% SD, p = 0.007).

This does not alter the main message of Figure 6, which is that the response correlations across concentrations in concentration-invariant subpopulations are strikingly different from those of the entire ensemble, which strongly decorrelate with increasing concentrations (Figure 5). We agree that there might be small differences with some odors between the representations of low and high concentrations. Crucially, these differences are too small to make the responses for low and high concentration discriminable on a single trial basis, as we show in Figure 6. We have included a more detailed description of these data in Results section of the revised manuscript (subsection “A concentration-invariant subpopulation of piriform neurons”, second paragraph, and Figure 6—figure supplement 1).

*2) The small fraction of cells that are classified as concentration invariant, remains a concern, as well as lack of evidence that these cells in the data set indeed constitute a cell type as defined by layer, specific input/output patterns or any other features, besides the difference in observed responses (though proposed as such in the manuscript). It is also unclear what the decoding scheme is for a wider concentration range, except an actual change in perceived odor identity mentioned in the text.*

*It is indeed important to document that a higher percentage of neurons are called as concentration invariant in the piriform cortex compared to the bulb. However, I'm not convinced that the strong message and title of the manuscript should be focused on a result that summarizes, in the best case scenario, the behavior of ~10% of the responsive neurons in the absence of any additional evidence that these 10% of responsive neurons are doing the job as proposed.*

Given that only 10% of cells (out of ~1 million cells in piriform cortex) are identified as concentration-invariant and we have not yet identified additional identifying features of these cells, the reviewer is uncomfortable with us using a title that focuses exclusively on the concentration-invariant subpopulation. In fact, we agree that this is just one of multiple novel results that we present in this manuscript. Please note we that here we also provide for the first time, a detailed analysis of trial-to-trial variability within and across concentration for the general population, quantify normalization of piriform responses across concentrations, and demonstrate that there is no discernible topography in piriform representations. We have therefore made the title of this manuscript more generic (“Odor identity coding by distributed ensembles of neurons in the mouse olfactory cortex”), per reviewer 3’s suggestion.

*3) The anesthetized vs. awake explanation is not robust. In my opinion, different cells can be differentially affected by the brain state, depending on the local and long range inputs they receive and the strength of corresponding activity patterns.*

We agree with the reviewer that different piriform cell types could differentially be affected by anesthesia. We have extended our discussion of potential anesthesia effects to include this point (subsection “Comparison with extracellular recordings in awake mice”, first and last paragraphs).